# Rethinking Classifier Re-Training in Long-Tailed Recognition: Label Over-Smooth Can Balance

**Siyu Sun**[1,2#], **Han Lu**[1,2#], **Jiangtong Li**[3], **Yichen Xie**[4], **Tianjiao Li**[2]
**Xiaokang Yang**[1], **Liqing Zhang**[1], **Junchi Yan**[1*]
[1]Department of CSE & MoE Key Lab of AI, Shanghai Jiao Tong University
[2]Bilibili Inc    [3]Tongji University    [4]UC Berkeley
{sunsiyu, sjtu_luhan, xkyang, yanjunchi}@sjtu.edu.cn
jiangtongli@tongji.edu.cn, zhang-lq@cs.sjtu.edu.cn
yichen_xie@berkeley.edu, litianjiao01@bilibili.com
Code: https://github.com/Thinklab-SJTU/LOS

## Abstract

In the field of long-tailed recognition, the Decoupled Training paradigm has shown exceptional promise by dividing training into two stages: representation learning and classifier re-training. While previous work has tried to improve both stages simultaneously, this complicates isolating the effect of classifier re-training. Recent studies reveal that simple regularization can produce strong feature representations, highlighting the need to reassess classifier re-training methods. In this study, we revisit classifier re-training methods based on a unified feature representation and re-evaluate their performances. We propose two new metrics, Logits Magnitude and Regularized Standard Deviation, to compare the differences and similarities between various methods. Using these two newly proposed metrics, we demonstrate that when the Logits Magnitude across classes is nearly balanced, further reducing its overall value can effectively decrease errors and disturbances during training, leading to better model performance. Based on our analysis using these metrics, we observe that adjusting the logits could improve model performance, leading us to develop a simple label over-smoothing approach to adjust the logits without requiring prior knowledge of class distribution. This method softens the original one-hot labels by assigning a probability slightly higher than $\frac{1}{K}$ to the true class and slightly lower than $\frac{1}{K}$ to the other classes, where $K$ is the number of classes. Our method achieves state-of-the-art performance on various imbalanced datasets, including CIFAR100-LT, ImageNet-LT, and iNaturalist2018.

## 1 Introduction

Real-world datasets often exhibit long-tailed distributions, where the majority of samples belong to a few major classes, while the remaining samples are spread across many rare classes (Buda et al., 2018; Zhang et al., 2023). Although traditional classification methods (He et al., 2016; Redmon & Farhadi, 2017) perform well on balanced datasets, they may not be effective on imbalanced datasets as they tend to prioritize majority classes and neglect minority ones (Buda et al., 2018; He & Garcia, 2009; Van Horn & Perona, 2017). Long-tail recognition (LTR) has emerged as a challenging and important task, where models are trained on imbalanced datasets and are expected to perform well across both majority and minority classes (Zhang et al., 2023).

Various techniques are developed for LTR, including class re-weighting (Cui et al., 2019; Lin et al., 2017; Ren et al., 2020), class re-sampling (Kang et al., 2019; Wang et al., 2020a), and others. Among these techniques, the Decoupled Training paradigm (Kang et al., 2019) shows remarkable efficacy.

---

*Han Lu and Siyu Sun contribute equally. Junchi Yan is the correspondence author. Work was in part supported by NSFC (62222607, 62402341), Shanghai Municipal Science and Technology Major Project (2021SHZDZX0102), and the Postdoctoral Fellowship Program of CPSF (GZC20241225).

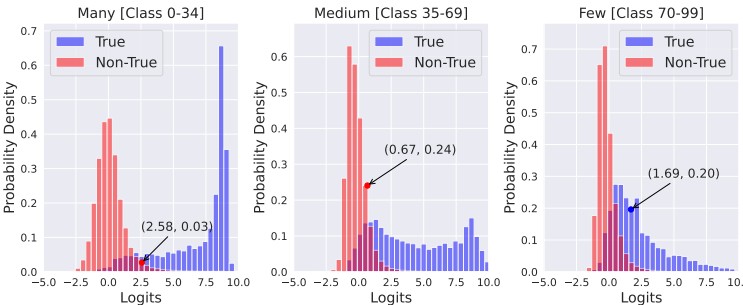

(a) **Vanilla Cross-Entropy (CE).** "Many" class logit 2.58 is larger than the ground truth "Few" class logit 1.69, leading to the misclassification.

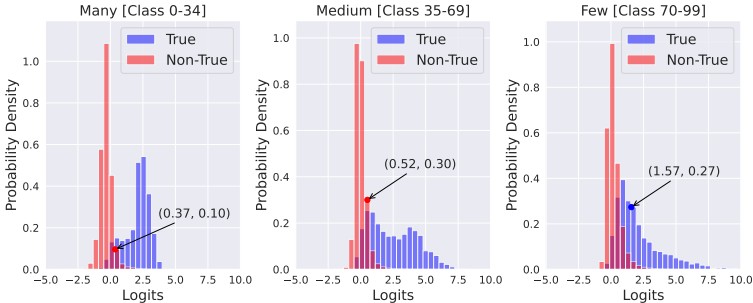

(b) **LOS (ours).** "Few" class logit 1.57 outperforms the other classes.

Figure 1: **An Intuitive View of Logits Magnitude Influence.** The two models are trained on CIFAR100-LT with IR=100. The blue columns represent the logits of the true samples of current class, while the red columns correspond to the logits of the non-true samples. We take the instance indexed 7202 in test set as an example (the point). Its true class is 75, but it is erroneously classified as class 3 in Many class by Vanilla CE . Our method LOS can obtain the more balanced and smaller logits magnitudes (defined for each class as the difference between the mean logits assigned to the true class and the mean logits assigned to the other non-true classes), helping discrimination in a multi-class classification scenario.

This approach divides the training process into two different stages: representation learning and classifier retraining. In the first stage, the model is trained to learn a general feature representation without considering class imbalance. In the second stage, the classifier is retrained using the acquired feature representation, employing techniques to address class imbalance.

Previous works (Alshammari et al., 2022; Desai et al., 2021; Ren et al., 2020; Zhong et al., 2021) on Decoupled Training often improve both representation learning and classifier retraining simultaneously, which can obscure the individual impact of classifier retraining. LTWB (Alshammari et al., 2022) shows that simple regularization techniques can outperform the original representation learning methods. These findings highlight the need to evaluate classifier retraining separately to isolate its effects and understand its true contribution to model performance. By using unified feature representations, we can better identify effective factors and make improvements.

Therefore, we revisit classifier retraining methods, including re-weighting (Kang et al., 2019; Cao et al., 2019; Ren et al., 2020; Lin et al., 2017; Cui et al., 2019), re-sampling (Shen et al., 2016), parameter regularization (Kang et al., 2019; Alshammari et al., 2022), and direct logits modification during inference (Kang et al., 2019; Menon et al., 2021). For the fair comparison of these baselines, we design a unified formulation and use consistent and high-quality first-stage feature representations to conduct accuracy experiments. The performance of the re-evaluated methods, as shown in Tab. 1, far exceeds those claimed in previous literature, providing a new, fair benchmark for further analysis.

Based on our experimental results, we conduct further analysis. Previous works (Alshammari et al., 2022; Kang et al., 2019) use classifier weight norm to show that their models do not overfit majority classes, but this metric is unreliable as it ignores feature magnitude and limits representational capacity. We propose a more reliable metric: Logits Magnitude (LoMa), defined for each class as the difference between the mean logits assigned to the true class and the mean logits assigned to

other classes. We observe that performance correlates positively when the LoMa is balanced across all classes. Since directly optimizing LoMa is impractical, we introduce the Regularized Standard Deviation, calculated as the standard deviation of the logits divided by the LoMa. By reducing the LoMa, particularly when it is balanced across classes, we can minimize training disturbances and mitigate the effects of class imbalance, which inspired the development of our method.

We propose "Label Over-Smooth" (LOS), a simple approach similar in form to label smoothing (Huang et al., 2019) that reduces LoMa while maintaining class balance without needing prior knowledge of class distribution. Our LOS method softens the original one-hot labels by assigning a probability slightly higher than $\frac{1}{K}$ to the true class and slightly lower than $\frac{1}{K}$ to the other classes, where $K$ is the number of classes. By over-smoothing the labels, LOS effectively reduces the LoMa and maintains class balance. Both theoretically and experimentally, we show that higher non-true label probabilities improve results. As illustrated in Fig. 1, LOS maintains the discriminative power between the true class and other classes within each sample. It reduces the influence of dominant classes and diminishes majority class dominance, enhancing sample discernibility. Our method achieves state-of-the-art performance on various mainstream imbalanced datasets, such as CIFAR100-LT, ImageNet-LT, and iNaturalist2018. Extensive ablation studies also verify the robustness of our method to hyperparameter variations. Additionally, LOS can serve as a plug-and-play retraining method that effectively enhances the performance of existing classifiers.

Our contributions can be summarized as follows:
- We revisit previous classifier retraining methods for long-tailed recognition, re-evaluating them under a unified framework to enable fair comparisons and further analysis.
- We propose two metrics, Logits Magnitude (LoMa) and Regularized Standard Deviation, to effectively compare the distributions of logits values across different methods and classes. These metrics provide insights into performance and highlight conditions for achieving improved results.
- We introduce a simple Label Over-Smoothing approach (LOS) that does not require prior knowledge of class distribution. LOS mitigates overfitting and reduces bias caused by class imbalance in the training set, achieving state-of-the-art performance on three commonly used datasets.

## 2 RETHINKING CLASSIFIER RE-TRAINING

### 2.1 REVISITING CLASSIFIER RE-TRAINING METHODS

In long-tailed recognition, the Decoupled Training Paradigm (Kang et al., 2019) can effectively improve model performance. Various training paradigms for classification heads have been explored, including both joint training from scratch (Cao et al., 2019; Ren et al., 2020; Shen et al., 2016) and fine-tuning only (Kang et al., 2019; Lin et al., 2017; Cui et al., 2019; Alshammari et al., 2022; Menon et al., 2021). However, there is still a lack of systematic studies to determine the most appropriate method for classifier retraining, as previous works (Alshammari et al., 2022; Cui et al., 2019; Ren et al., 2020; Zhong et al., 2021) often use different backbone feature extractors, making comparisons difficult. To provide a fair comparison, we conduct an analysis of these methods using the same strong feature representations trained in the first stage by LTWB (Alshammari et al., 2022). During the retraining process, we freeze the weights of the backbone and train only the classification head.

**Notation.** Suppose there are $K$ classes with $n_i$ denoting the sample number of class $i$. The deep feature extracted from an image $\mathbf{x}$ is represented by $f(\mathbf{x})$, while the classifier weight vectors are denoted as $\boldsymbol{W} = [\boldsymbol{w}_1, ..., \boldsymbol{w}_K]$.

Inspired by (Wu et al., 2021), we construct a unified formula for the different methods:

$$\mathcal{L}(\boldsymbol{W}; f(\mathbf{x}), y) = -r_w(y) \cdot \log \left( \frac{e^{\mathbf{z}_y}}{\sum_i e^{\mathbf{z}_i}} \right),$$

$$\text{where } \mathbf{z}_i = g_i \left( \boldsymbol{w}_i, f(\mathbf{x}) \right). \tag{1}$$

Given input feature $f(\mathbf{x})$ and the label $y$, $r_w(y)$ denotes the re-weighting factor for current label categories and $\mathbf{z}_i = g_i(\boldsymbol{w}_i, f(\mathbf{x}))$ represents the calculated logit after classifier. The common used classifiers like Vanilla FC are $g(\boldsymbol{W}, f(\mathbf{x})) = \boldsymbol{W}^\top f(\mathbf{x}) + \boldsymbol{b}$. For simplicity, we omit the bias $\boldsymbol{b}$ in the table. The methods Vanilla Cosine, Learnable Weight Scaling (LWS) (Kang et al., 2019), Label-Distribution-Aware Margin Loss (Cao et al., 2019), Balanced-Softmax Loss (Ren et al., 2020) focus

Table 1: Study of the classifier re-training methods in decoupled learning scheme in LTR. Vanilla FC trains a general representation, performing itself in a role as both the baseline and the backbone feature extractor. Experiments are conducted on CIFAR100-LT with imbalanced ratio 100 and constant $\gamma, \beta, \delta, \tau$. See details for the listed formulation in Sec. 2.1.

| Methods | Formulation | All | Many | Medium | Few |
|---|---|---|---|---|---|
| Vanilla FC | $g_i = \boldsymbol{W}_i^\top f(\mathbf{x})$ | 48.44 | **78.74** | 47.37 | 14.33 |
| Vanilla Cosine | $g_i = \boldsymbol{W}_i^\top / \|\boldsymbol{W}_i^\top\| \cdot f(\mathbf{x})/\|f(\mathbf{x})\| \cdot 1/\tau$ | 52.20 | 77.21 | 50.52 | 24.98 |
| Learnable Weight Scaling (Kang et al., 2019) | $g_i = \mathbf{c}_i \cdot \boldsymbol{W}_i^\top f(x)$, where $\mathbf{c}_i$ is learnable | 48.99 | 78.51 | 47.51 | 16.26 |
| LDAM Loss (Cao et al., 2019) | $g_i = \boldsymbol{W}_i^\top f(\mathbf{x}) - \mathbf{1}\{i = y\} \cdot C/n_i^\gamma$ | 52.94 | 72.37 | **54.31** | 28.67 |
| Balanced-Softmax Loss (Ren et al., 2020) | $g_i = \boldsymbol{W}_i^\top f(\mathbf{x}) + \log(n_i)$ | 52.23 | 77.63 | 50.74 | 24.33 |
| Re-Sampling (Shen et al., 2016) | $r_s(i) \propto 1/n_i$, $r_s(i)$ is the sample ratio of class $i$ | 48.44 | 78.74 | 47.37 | 14.33 |
| Focal Loss (Lin et al., 2017) | $r(y) = (1 - \mathbf{p}_y)^\gamma$, $\mathbf{p}_y$ is the predicted probability of $y$ | 50.94 | 78.00 | 49.97 | 20.50 |
| Class-Balanced Loss (Cui et al., 2019) | $r(y) = (1 - \beta)/\left(1 - \beta_y^n\right)$ applied with BCE loss | **53.31** | 70.88 | 50.25 | **36.41** |
| MaxNorm (Alshammari et al., 2022) | $\|\boldsymbol{W}_i\|_2^2 \le \delta^2$ | 51.64 | 77.60 | 49.37 | 23.99 |
| $\tau$-normalized (Kang et al., 2019) | $h_i = \left(\boldsymbol{W}_i/\|\boldsymbol{W}_i\|^\tau\right)^\top f(\mathbf{x})$ | 53.01 | 74.74 | 47.31 | 34.30 |
| Post-hoc Logit Adjustment (Menon et al., 2021) | $h_i = \boldsymbol{W}_i^\top f(\mathbf{x}) - \tau \log(n_i)$ | 52.45 | 75.57 | 47.71 | 31.00 |

on designing different classifier function $g$. Re-Sampling (Shen et al., 2016) guarantees the same sampling probability of each class and we use $r_s(i)$ as the sampling ratio, acting as the supplementary of the equation. Focal Loss (Lin et al., 2017) and Class-Balanced Loss (Cui et al., 2019) design the class weights $r_w(y)$ according to class label $y$. In addition to the general formulation in Eq. 1, there are other perspectives. MaxNorm (Alshammari et al., 2022) regularizes the weight norm through Projected Gradient Descent (PGD) in practice. $\tau$-normalized (Kang et al., 2019) and Logit Adjusted Loss (Menon et al., 2021) use $h_i$ as the logit modification function in the inference stage. We conduct experiments on CIFAR100-LT dataset with imbalanced ratio 100 and report the results in Tab. 1.

## 2.2 INSIGHTS

Before delving into these methods, we will first present several propositions that aim to offer intuitive insights and foundational understanding relevant to the Long-tail problem.

Consider the general form of CE without regularization, assuming the label vector as $\mathbf{y}$ instead of one-hot label in Eq. 1.

$$\mathcal{L}(\boldsymbol{W}, \boldsymbol{b}; f(\mathbf{x}), \mathbf{y}) = \sum_i -\mathbf{y}_i \cdot \log\left(\frac{e^{\mathbf{z}_i}}{\sum_j e^{\mathbf{z}_j}}\right), \tag{2}$$

where $\mathbf{z} = \boldsymbol{W}^\top f(\mathbf{x}) + \boldsymbol{b}$ and define $\mathbf{s}$ as the final predicted probability that $\mathbf{s}_i = e^{\mathbf{z}_i}/\sum_j e^{\mathbf{z}_j}$.

**Proposition 1 (Bias Convergence).** *Given a deterministic matrix $\boldsymbol{W}$, loss function $\mathcal{L}$ is a convex function with respect to $\boldsymbol{b}$ and will reach the global minimum as the corresponding bias $\boldsymbol{b}$ converges.*

*Proof.* Calculate the Hessian matrix of $\mathcal{L}$ w.r.t. $\boldsymbol{b}$ as $\boldsymbol{H}$

$$\boldsymbol{H}_{i,j} = \begin{cases} \mathbf{s}_i(1 - \mathbf{s}_i) & \text{if } i = j \\ -\mathbf{s}_i \mathbf{s}_j & \text{if } i \ne j \end{cases} \tag{3}$$

The Hessian matrix $\boldsymbol{H}$ is a positive semi-definite matrix because $\boldsymbol{x}\boldsymbol{H}\boldsymbol{x}^\top \ge 0$ for any $\boldsymbol{x}$. (Refer to detailed proof in Appendix B). $\square$

Therefore, our focus can shift to the optimization problem concerning the weight matrix $\boldsymbol{W}$. In all cross-entropy based methods, the aforementioned conclusion remains valid.

**Proposition 2 (Arbitrary Length).** *Take the basic cross entropy as the loss function, as we defined in Eq. 2. If there exist optimal parameters $\boldsymbol{W}^*$ and $\boldsymbol{b}^*$ such that convergence is achieved, then there exist a series of convergence points $(\boldsymbol{W}', \boldsymbol{b}^*)$ where $\boldsymbol{W}'$ can have arbitrary magnitudes.*

*Let $\boldsymbol{w}_i' = \boldsymbol{w}_i^* + \delta$, we have*

$$s_i' = \frac{e^{\mathbf{z}_i'}}{\sum_j e^{\mathbf{z}_j'}} = \frac{1}{\sum_j e^{\boldsymbol{W}_j' f(\mathbf{x}) + \boldsymbol{b}_j^* - \boldsymbol{W}_i' f(\mathbf{x}) - \boldsymbol{b}_i^*}} = \frac{1}{\sum_j e^{\mathbf{z}_j - \mathbf{z}_i}} = \mathbf{s}_i, \tag{4}$$

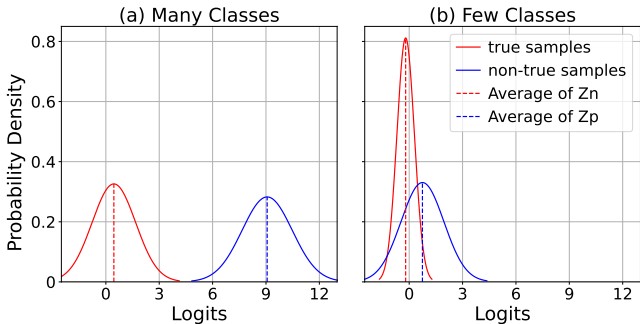

Figure 2: **Visual interpretation of Logits Magnitude and Standard Deviation.** The LoMa **L** is defined as the difference between the mean logits of true class (dashed blue line) and other non-true classes (dashed red line). The distribution simulates a normal distribution.

$$\mathcal{L}(\boldsymbol{W}', \boldsymbol{b}^*; f(\mathbf{x}), y) = \mathcal{L}(\boldsymbol{W}^*, \boldsymbol{b}^*; f(\mathbf{x}), y) \tag{5}$$

*The predicted probability is not affected by $\delta$ as the length $||\boldsymbol{w}_i||_2$ of each $\boldsymbol{w}_i$ could be changed.*

Proposition 2 highlights that the magnitude of weight vectors can be arbitrary without affecting optimal prediction. This suggests that **directly comparing $||w_i||_2^2$ across classes is meaningless. To address this, we propose Logits Magnitude, which enables unified comparision across classes.**

### 2.3 LOGITS MAGNITUDE AND REGULARIZED STANDARD DEVIATION

Previous techniques such as MaxNorm (Alshammari et al., 2022) and $\tau$-Norm (Kang et al., 2019) have been employed to constrain the length of each vector in $\boldsymbol{W}$. However, these Weight Norm constraint methods cannot effectively characterize the relationships between different classes. Therefore, we propose a more reasonable metric directly from the perspective of logits and compare it with the commonly used Weight Norm as follows:

**Definition 1** (**Logits Magnitude**). *Suppose the number of classes is $K$, $\mathbf{z}_{Pi}$ represents the logits of samples of class $i$ and $\mathbf{z}_{Ni}$ represents the logits of the samples for non-true classes (excluding class $i$). For each true class, the LoMa $\mathbf{L} \in \mathbb{R}^K$ is defined as the difference between the mean logits of true class and other classes.*

$$\mathbf{L_i} = \mathbb{E}[\mathbf{z}_{Pi}] - \mathbb{E}[\mathbf{z}_{Ni}]. \tag{6}$$

Fig. 2 provides an intuitive visual interpretation of LoMa. We compute the mean and variance of the logits for the true and non-true classes, and fit them to a normal distribution. We display schematic diagrams for the "Many" and "Few" classes as a whole for comparison.

LoMa is effective in characterizing the performance. As observed in Fig. 3a where the methods are listed in increasing order of performance, **we have empirically found that achieving a more uniform distribution of LoMa tends to yield improved performance**. Fig. 1 illustrates the intuitive advantage of balanced LoMa on performance, particularly for the few classes.

Previous methods such as (Alshammari et al., 2022; Kang et al., 2019) often incorporate regularization as a crucial parameter. Regularization helps align each $\mathbf{L_i}$, but decreases the capabilities of $\boldsymbol{W}$. LoMa shows reduced bias susceptibility and a strong correlation with classification accuracy, unlike vector length magnitudes. However, it is challenging to optimize directly during training. To enhance performance under balanced LoMa, we introduce Regularized Standard Deviation, a new metric to evaluate and overcome these challenges.

**Definition 2** (**Regularized Standard Deviation**). *Given the logits $\mathbf{z}$, the standard deviation $\sigma(\mathbf{z})$ and logits magnitude $\mathbf{L}$. The regularized standard deviation $\mathbf{R} \in \mathbb{R}^K$ is defined as:*

$$\mathbf{R}_i = \frac{\sigma(\mathbf{z}_i)}{\mathbf{L}_i}. \tag{7}$$

The corresponding $\mathbf{R}_i$ increases gradually as the classes transition from head to tail shown in Fig. 3b. This observation is in line with general intuition, as classes with more training samples typically

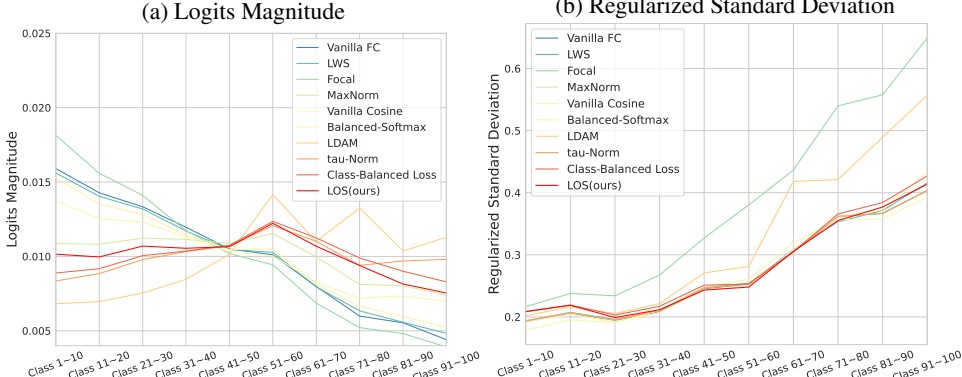

Figure 3: **Overview of our proposed metrics. Left: Overview of Logits Magnitude** for various methods on CIFAR100-LT with an imbalanced ratio of 100. Classes are grouped into segments of 10, and mean values are computed for comparative analysis. **The methods in the legend are sorted in ascending order of performance from top to bottom.** The difference in means between samples from true and non-true classes is evaluated for each class in the test set. Magnitude regularization using the 1-norm is employed to enhance comparability. **Right: Overview of Regularized Standard Deviation**. The true distribution of logits **z** for each class is not available considering the lack of the samples, so the displayed results represent computations on the test set.

exhibit better feature extraction. Importantly, the computed $\mathbf{R}_i$ values are quite similar for models derived using different optimization techniques. **Therefore, the Regularized Standard Deviation can serve as an approximate invariant for further analysis.**

In conclusion, we introduce LoMa as an effective metric for characterizing model performance, demonstrating that a more uniform distribution correlates with improved outcomes, particularly for minority classes. Additionally, Regularized Standard Deviation exhibits minimal variation across different models, with consistent trends.

## 3 METHOD

### 3.1 DEEP DIVE INTO LOGITS MAGNITUDE (LoMA)

In classification problems, minimizing the loss $\mathcal{L}$ and maximizing classification accuracy are not completely equivalent. Based on the above observation, we find that the computation of gradients for parameters associated with the "Few" classes is significantly influenced by sampling and process perturbations, whereas the impact on "Many" classes is relatively mild. This manifests as the model often being overconfident in the "Many" classes while exhibiting unstable predictions for the "Few" classes. We discuss this process in the following.

Suppose we achieve relatively balanced magnitudes for each class, as observed in Sec. 2.3, denoted as $\mathbf{L}_1 \approx \mathbf{L}_2 \approx ... \approx \mathbf{L}_K$. Despite this balance, it is inevitable to encounter variations in the standard deviations $\sigma(\mathbf{z}_i) = \mathbf{R}_i\mathbf{L}_i$ across different classes. These variations significantly impact the training dynamics and the computation of gradients during the training process. **Although it is not feasible to completely eliminate the disparity in $\mathbf{R_i}\mathbf{L_i}$ caused by the data distribution, the impact can be mitigated by significantly reducing the magnitudes L.**

Consider random perturbations $\Delta_i$ that independently affect each class $\mathbf{z}_i = w_i f_i + b_i$ and are proportional to the standard deviation of their logits, i.e., $\mathbf{z}'_i = \mathbf{z}_i + \Delta_i$ and the perturbation $\Delta_i \sim \mathbf{R}_i\mathbf{L}_i\epsilon$. Here, $\epsilon$ is a random variable with an expected value of $\mathbb{E}[\epsilon] = 0$. By modeling these perturbations, we can better understand their impact on training dynamics and develop strategies to mitigate adverse effects. These perturbations $\Delta$ are introduced during the training process, particularly due to factors such as overfitting in the initial stage and bias in the sampling of the training set.

**Proposition 3.** *When comparing the use of $\mathbf{z}_i$ and $\mathbf{z}'_i$ for subsequent calculations, the impact is minimal for balanced datasets but can be significant for imbalanced datasets.*

The expected value of probability $\mathbf{s}'_i$ can be expressed as:

$$\mathbb{E}[\mathbf{s}'_i] = \mathbb{E}\left[\frac{1}{\sum_j e^{\mathbf{z}'_j - \mathbf{z}'_i}}\right] = \mathbb{E}\left[\frac{1}{\sum_j e^{(\Delta_j - \Delta_i)} e^{(\mathbf{z}_j - \mathbf{z}_i)}}\right]. \tag{8}$$

In the balanced datasets, expected with $\mathbf{R}_1 \approx \mathbf{R}_2 \approx \cdots \approx \mathbf{R}_K$, $(\mathbf{R}_j \mathbf{L}_j - \mathbf{R}_i \mathbf{L}_i)$ is close to zero.

$$\mathbb{E}[\mathbf{s}'_i] \approx \frac{1}{\mathbb{E}[\exp(\Delta_j - \Delta_i)]} \mathbb{E}\left[\frac{1}{\sum_j e^{\mathbf{z}_j - \mathbf{z}_i}}\right] = \frac{\mathbb{E}[\mathbf{s}_i]}{\mathbb{E}[\exp(\Delta_j - \Delta_i)]} \tag{9}$$

where $\mathbb{E}[\exp(\Delta_j - \Delta_i)]$ can be considered as a constant $\mathcal{E}$. See the Appendix C for a detailed analysis of the approximation. For balanced datasets, the impact of disturbances is consistent across different classes $i$.

$$\mathbb{E}[\mathbf{s}'_i] / \mathbb{E}[\mathbf{s}_i] \approx 1/\mathcal{E}. \tag{10}$$

However, for imbalanced datasets, Eq. 9 no longer holds. There is a significant disparity in the values of $\exp(\Delta_j - \Delta_i)$ for different class $j$. If $\mathbf{R}_j < \mathbf{R}_k$, $\mathbb{E}[\exp(\Delta_j - \Delta_i)] < \mathbb{E}[\exp(\Delta_k - \Delta_i)]$. For instance, when $\epsilon$ follows a normal distribution we have

$$\mathbb{E}[\exp(\Delta_j - \Delta_i)] = \exp((\sigma(\Delta_j - \Delta_i))^2/2), \tag{11}$$

where $\sigma(\Delta_j - \Delta_i)$ represents the standard deviations of the random variable $(\Delta_j - \Delta_i)$. As a result, the impact of perturbations is directly correlated with the differences in standard deviations between classes, leading to $\mathbb{E}[\mathbf{s}'_i] / \mathbb{E}[\mathbf{s}_i]$ not consistent across classes $i$, where $\mathbb{E}[\mathbf{s}'_i] / \mathbb{E}[\mathbf{s}_i] \neq 1/\mathcal{E}$. The expectation $\mathbb{E}[\mathbf{s}_i]$ plays a crucial role as it not only relates to the final predictions but also has a direct correlation with the derivatives obtained from the gradient descent process. Specifically, the expectation of the derivative $\mathbb{E}[\frac{\partial \mathcal{L}}{\partial \mathbf{z_i}}] = \mathbb{E}[(\mathbf{s}_i - \mathbf{y}_i)]$ can be computed as the difference between the expectations $\mathbb{E}[\mathbf{s}_i]$ and $\mathbb{E}[\mathbf{y}_i]$.

Therefore, in the long-tailed scenario, the perturbations $\Delta$ introduced in training process will have a significant impact on model training, ultimately compromising the final classification accuracy.

**Deliberately reducing the logits magnitudes holds promise for addressing the effects caused by these perturbations.** Previous methods (Alshammari et al., 2022; Kang et al., 2019) can achieve implicit reduction of LoMa by regularizing the weights of the prediction head. However, they also limit the expressive power of the classification head. Other methods (Cao et al., 2019; Ren et al., 2020; Cui et al., 2019; Menon et al., 2021) also fail to recognize the importance of deliberately optimizing this effective objective. To overcome these limitations, we propose "Label Over-Smoothing" approach (LOS) to soften the original one-hot labels and achieve balanced logits magnitude.

## 3.2 LABEL OVER-SMOOTH

Based on the above analysis, we can observe that when the value of $\mathbf{s}_i$ is larger, the impact of perturbation is smaller, and the differences between classes are also smaller. So, an intuitive idea is to make $\mathbf{s}_i$ as large as possible during the process.

In our method, we softens the original one-hot labels by assigning a probability slightly higher than $\frac{1}{K}$ to the true class and slightly lower than $\frac{1}{K}$ to the other classes, where $K$ is the number of classes. It is essential to ensure that the probability of the true class is higher than the probability of each non-true class for model convergence and discriminability in the true class.

Suppose the number of the categories is $K$. Given the input image $\mathbf{x}$ and label $y$, feature extractor backbone $f(\cdot)$ and classifier $\theta = (\boldsymbol{W}, \boldsymbol{b})$, the formulation of our "Label Over-Smooth" is as follows:

$$\mathcal{L}(\theta; f(\mathbf{x}), y) = \sum_{i=1}^{K} -\tilde{\mathbf{y}}_i \cdot \log\left(\frac{e^{\mathbf{z}_i}}{\sum_j e^{\mathbf{z}_j}}\right),$$

$$\tilde{\mathbf{y}}_i = \begin{cases} 1 - \delta + \frac{\delta}{K}, & i = y \\ \frac{\delta}{K}, & i \neq y \end{cases} \tag{12}$$

where $\delta \in [0, 1)$ is a constant value to control the overall non-true class probability. Interestingly the above formulation is fundamentally consistent with the conventional Label Smoothing

Table 2: **Comparison for CIFAR100-LT, ImageNet-LT and iNaturalist2018 Benchmarks.** Top-1 accuracy (%) is reported and CIFAR100-LT consists of three imbalanced ratio (IR) 100/50/10. Our model outperforms all the baselines, showing satisfactory accuracy in the few-class categories.

| Method | Reference | CIFAR100-LT | | | ImageNet-LT | | | | iNaturalist2018 | | | |
|---|---|---|---|---|---|---|---|---|---|---|---|---|
| | | IR=100 | IR=50 | IR=10 | Many | Medium | Few | All | Many | Medium | Few | All |
| CE | — | 38.3 | 43.8 | 55.7 | **65.9** | 37.5 | 7.7 | 44.4 | **72.2** | 63.0 | 57.2 | 61.7 |
| Focal (Lin et al., 2017) | ICCV17 | 38.4 | 44.3 | 55.7 | 36.4 | 29.9 | 16.0 | 30.5 | — | — | — | 61.1 |
| LDAM-DRW (Cao et al., 2019) | NIPS19 | 42.0 | 46.6 | 58.7 | — | — | — | 48.8 | — | — | — | — |
| cRT (Kang et al., 2019) | ICLR19 | — | — | — | 61.8 | 46.2 | 27.3 | 49.6 | 69.0 | 66.0 | 63.2 | 65.2 |
| $\tau$-norm (Kang et al., 2019) | ICLR19 | 47.7 | 52.5 | 63.8 | 59.1 | 46.9 | 30.7 | 49.4 | 65.6 | 65.3 | 65.5 | 65.6 |
| CB-CE (Cui et al., 2019) | CVPR19 | 39.6 | 45.3 | 58.0 | 39.6 | 32.7 | 16.8 | 33.2 | 53.4 | 54.8 | 53.2 | 54.0 |
| De-confound(Tang et al., 2020) | NIPS20 | 44.1 | 50.3 | 59.6 | 62.7 | 48.8 | 31.6 | 51.8 | — | — | — | — |
| BALMS(Ren et al., 2020) | NIPS20 | 50.8 | — | 63.0 | 61.1 | 48.5 | 31.8 | 50.9 | 65.5 | 67.5 | 67.5 | 67.2 |
| BBN (Zhou et al., 2020) | CVPR20 | 42.6 | 47.0 | 59.1 | — | — | — | — | 49.4 | 70.8 | 65.3 | 66.3 |
| LogitAjust(Menon et al., 2021) | ICLR21 | 42.0 | 47.0 | 57.7 | — | — | — | 51.1 | — | — | — | 69.4 |
| DisAlign (Zhang et al., 2021a) | CVPR21 | — | — | — | 61.3 | 52.2 | 31.4 | 52.9 | 69.0 | **71.1** | — | 70.6 |
| LTWB (Alshammari et al., 2022) | CVPR22 | 53.4 | 57.7 | 68.7 | 62.5 | 50.4 | 41.5 | 53.9 | 71.2 | 70.4 | 69.7 | 70.2 |
| AREA (Chen et al., 2023) | ICCV23 | 48.8 | 51.8 | 60.1 | — | — | — | 49.5 | — | — | — | 68.4 |
| RBL (Peifeng et al., 2023) | ICML23 | 53.1 | 57.2 | — | 64.8 | 49.6 | 34.2 | 53.3 | — | — | — | — |
| EWB-FDR (Hasegawa & Sato, 2024) | ICLR24 | 53.0 | — | — | 63.4 | 50.0 | 35.1 | 53.2 | — | — | — | — |
| **LOS (ours)** | | **54.9** | **58.8** | **69.7** | 63.2 | 50.7 | 42.3 | 54.4 | 69.2 | 70.7 | **71.3** | **70.8** |

approach (Szegedy et al., 2016) and we can name the hyper-parameter $\delta$ as "Smooth Factor". **The primary distinction lies in the $\delta$, which can be set as a quite large value in our approach**.

To our best knowledge, Label Smoothing is typically employed in a lightweight manner. For example, $\delta$ is usually set as $0.2$. Our method represents a novel approach in the LTR field, setting $\delta$ as $0.98$ or $0.99$ to achieve SOTA results. As theoretically analysis in Sec. 3.1, a larger label smoothing value can reduce the LoMa to achieve better performance. From this perspective, $\delta$ approaching 1 can yield the best results. However, our method essentially acts as an auxiliary regularization technique to mitigate biases among different class samples and assist the model in converging. The model needs to exhibit both convergence and discriminability, meaning that the ground truth labels should remain distinct from the other labels. We believe this represents a trade-off, and we demonstrate this phenomenon experimentally in Sec. 4.3.

Compared to previous methods, LOS directly focus on optimizing the fundamental objective LoMa. Our method not only achieves outstanding experimental performance but also offers ease of implementation and independence from prior information about class distribution. This represents a significant improvement in both theoretical analysis and empirical performance over existing methods.

## 4 EXPERIMENTS

We conduct experiments on long-tailed image classification benchmarks: CIFAR100-LT (Krizhevsky et al., 2009), ImageNet-LT (Liu et al., 2019) and iNaturalist2018 (Van Horn et al., 2018). All experiments are implemented using PyTorch and run on GeForce RTX 3090 (24GB) and A100-PCIE (40GB). Source code will be made publicly available.

### 4.1 DATASETS AND IMPLEMENTATION DETAILS

We define the class number as $K$ and imbalanced ratio (IR) of the long-tailed datasets as IR = $n_{max}/n_{min}$, where $n_{max}$ and $n_{min}$ is the number of training samples in the largest and smallest class, respectively. In line with the previous SOTA work (Alshammari et al., 2022), we use ResNet34 (He et al., 2016) in CIFAR100-LT, ResNeXt50 (Xie et al., 2017) in ImageNet-LT and ResNet50 (He et al., 2016) in iNaturalist2018. For smooth factor, we use $0.98$ in CIFAR100-LT and $0.99$ in ImageNet-LT and iNaturalist2018. More details of the datasets and implementation are included in Appendix D.

**Evaluation Metrics.** We evaluate the models on corresponding balanced test dataset and report top-1 accuracy. In line with the protocol in literature (Liu et al., 2019), we give accuracy on three different splits of classes with varying numbers of training data: many (over 100 images), medium (20~100 images), and few (less than 20 images).

### 4.2 BENCHMARK RESULTS

**Compared Methods.** We selected the popular methods in LTR and split them into two categories. (i) Baseline models: Some of these models modify the loss directly, such as cross entropy (CE) loss, class

Table 3: **Plugin for CIFAR100-LT, ImageNet-LT.** Top-1 accuracy (%) is reported and CIFAR100-LT consists of three imbalanced ratio (IR) 100/50/10. Our method can be directly plugged into non-conflicting upperbound methods by finetuning the classifier and can significantly improve results.

| Method | Reference | CIFAR100-LT | | | ImageNet-LT | | | |
| | | IR=100 | IR=50 | IR=10 | Many | Medium | Few | All |
|---|---|---|---|---|---|---|---|---|
| **Vallina CE + LOS (ours)** | | **54.9** | **58.8** | **69.7** | 63.2 | **50.7** | **42.3** | **54.4** |
| RIDE (Wang et al., 2020b) | ICLR20 | 49.1 | — | — | 67.9 | 52.3 | 36.0 | 56.1 |
| SSD (Li et al., 2021) | ICCV21 | 46.0 | 50.5 | 62.3 | 66.8 | 53.1 | 35.4 | 56.0 |
| PaCo (Cui et al., 2021) | ICCV21 | 52.0 | 56.0 | 64.2 | 63.2 | 51.6 | 39.2 | 54.4 |
| BCL (Zhu et al., 2022) | CVPR22 | 51.0 | 54.9 | 64.4 | 65.3 | 53.5 | 36.3 | 55.6 |
| NCL (Li et al., 2022a) | CVPR22 | 53.3 | 56.8 | — | — | — | — | 57.4 |
| GML (Suh & Seo, 2023) | ICML23 | 53.0 | 57.6 | 65.7 | 68.7 | 55.7 | 38.6 | 58.3 |
| ProCo (Du et al., 2024) | TPAMI24 | 52.8 | 57.1 | 65.5 | — | — | — | 58.0 |
| **RIDE+LOS(ours)** | | 50.2(+1.1) | — | — | 66.9(-1.0) | 53.2(+0.9) | 37.9(+1.9) | 56.5(+0.4) |
| **SSD+LOS(ours)** | | 47.3(+1.3) | 51.4(+0.9) | 63.1(+0.8) | 66.2(-0.6) | 53.8(+0.7) | 36.7(+1.3) | 56.3(+0.3) |
| **PaCo+LOS(ours)** | | 52.9(+0.9) | 56.8(+0.8) | 65.3(+1.1) | 62.7(-0.5) | 52.5(+0.9) | 41.2(+2.0) | 55.0(+0.6) |
| **BCL+LOS(ours)** | | 52.2(+1.2) | 56.0(+1.1) | 65.1(+0.7) | 64.7(-0.6) | 54.1(+0.8) | 28.1(+1.8) | 56.1(+0.5) |
| **NCL+LOS(ours)** | | 54.1(+0.8) | 57.5(+0.7) | — | — | — | — | 57.7(+0.3) |
| **GML+LOS(ours)** | | 54.0(+1.0) | 58.4(+0.8) | 66.6(+0.9) | 68.0(-0.7) | 55.7(+0.6) | 40.7(+2.1) | 58.7(+0.4) |
| **ProCo+LOS(ours)** | | 53.9(+1.1) | 58.0(+0.9) | 66.7(+1.2) | — | — | — | 58.5(+0.5) |

Table 4: **Improvement for different feature representation backbones** (in different rows) on CIFAR100-LT with IR=100. LOS outperforms all finetuning methods (across columns).

| Backbone Training | Classifier Finetune Methods | | | | | | |
| | Vanilla | CE | Focal | LDAM | BS | CB-BCE | LOS(ours) |
|---|---|---|---|---|---|---|---|
| CE | 41.59 | 41.67(+0.08) | 44.18(+2.59) | 43.87(+2.28) | 45.03(+3.44) | 45.01(+3.42) | **45.18(+3.59)** |
| CE-DRW | 42.38 | 42.38(+0.00) | 42.55(+0.17) | 42.41(+0.03) | 44.27(+1.89) | 44.39(+2.01) | **44.88(+2.50)** |
| LDAM-DRW | 44.43 | 44.45(+0.02) | 44.58(+0.15) | 44.47(+0.04) | 45.34(+0.91) | 45.52(+1.09) | **45.86(+1.43)** |
| BS | 45.90 | 45.91(+0.01) | 46.04(+0.14) | 45.90(+0.00) | 46.81(+0.91) | 46.76(+0.86) | **47.08(+1.28)** |

balanced CE loss (CB-CE) (Cui et al., 2019) or focal loss (Lin et al., 2017). RBL (Peifeng et al., 2023) focuses on the perspective of feature learning. De-confound (Tang et al., 2020), LogitAjust (Menon et al., 2021), BALMS (Ren et al., 2020), AREA (Chen et al., 2023) correct the classifier from different analysis aspects. BBN (Zhou et al., 2020), LDAM-DRW (Cao et al., 2019) leverage a learning rate annealing strategy to balance the feature learning process and the re-weighting procedure. The more straightforward method is decoupled learning (Kang et al., 2019), consisting of cRT (Kang et al., 2019), $\tau$-norm (Kang et al., 2019), DisAlign (Zhang et al., 2021a) and LTWB (Alshammari et al., 2022). Among them, LTWB is the most competitive model, whose main contribution is to use simple regularization to learn better robust features in the first stage. EWB-FDR (Hasegawa & Sato, 2024) delved into the success of LTWB by exploring Fisher's Discriminant Ratio in depth and streamlining the process, albeit at the expense of reduced performance. (ii) Upper Bound models: These models tend to make complex design or need more computational resources. They include self-supervised pretraining method PaCo (Cui et al., 2021), BCL (Zhu et al., 2022), GML (Suh & Seo, 2023), ProCo (Du et al., 2024), additional augmentation training data method OPeN (Zada et al., 2022), NCL (Li et al., 2022a), multi-experts ensemble method RIDE (Wang et al., 2020b), transfer learning method SSD (Li et al., 2021). We utilize the results reported in these original works.

**Results and Analysis.** Tab. 2 presents the results for long-tailed dataset recognition across three benchmarks. Our method LOS achieves state-of-the-art performance on all the benchmarks. Compared to the origin SOTA model LTWB (Alshammari et al., 2022) on CIFAR100-LT, LOS achieves an improvement of $1\% \sim 1.5\%$ in CIFAR100-LT dataset, $0.5\%$ in ImageNet-LT and $0.6\%$ in iNaturalist2018. In the long-tail tasks without bells and whistles, the improvement is significant. On ImageNet-LT, LOS achieves a significant improvement in classification accuracy for minority classes compared to previous methods. In Tab. 3, we present the performance of the Upperbound models. On the CIFAR100-LT dataset, our method LOS, significantly outperforms others, whereas on ImageNet, these methods take the lead. We attribute this phenomenon to a general gap in feature learning on more challenging datasets. In CIFAR100-LT, even simple models can overfit the imbalanced training set, making the main goal to ensure feature generalization and robustness, while fine-tuning the classifier to adapt better to balanced data. As a result, LOS excels over other methods. However, for larger datasets, there remains considerable room for improvement in the backbone network's feature representation. Hence, more advanced methods hold a clear advantage in this aspect. Consequently,

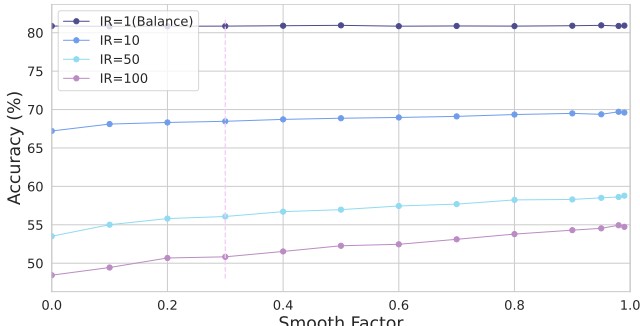

Figure 4: **Effect of Smooth Factor.** The overall accuracy will be improved with the growth of smooth factor. As the imbalance ratio (IR) increases, the improvement brought by our method becomes more significant. For fully balanced dataset (IR=1), our method has no effect. Moreover, it can be observed that the improvement between 0.2 and 0.3 is very gradual.

we leverage our method as a plugin to assist these methods in fine-tuning the classification head, yielding notable improvements, as shown in Tab. 3.

## 4.3 ABLATION STUDY

We further conduct ablation study on four aspects: 1) Effect of Smooth Factor, 2) Improvement for different backbone. See the Appendix E to view sensitivity of learning rate and weight decay.

**Effect of Smooth Factor.** The smooth factor is the only adjustable parameter in LOS. In Sec. 3.1, we suggest using a large label smooth factor, and further experiments exploring the impact of different values. We evaluate LOS using a range of smooth factors. The value 0.0 represents the Vanilla CE loss and the value 0.2 represents the common used label smoothing method (Szegedy et al., 2016) which uses soft labels to reduce overfitting. Due to the very gradual changes within the commonly used range of label smoothing, we believe this has led previous researchers to not pursue further investigation. Fig. 4 shows overall accuracy improvement with the increase of smooth factor. The best overall accuracy is achieved with 0.98 and the adjacent accuracy is close, within the allowable range of disturbance. **Therefore, we can choose the large smooth factor reassuringly.**

**Improvement for Different Backbone.** Our method has achieved excellent results based on strong feature representation, and we are interested in exploring the potential for improving models with arbitrary representations. Therefore, we first train traditional methods, freeze the trained backbone, and then finetune the classifier using different finetuning methods. In Tab. 4, our method exhibits the strongest ability to improve performance compared to the other fine-tuning methods evaluated. We find that the first-stage features without prior imbalanced information are more malleable, and different methods can obtain significant improvements, while imbalanced feature representations can not often be effectively improved. Interestingly, the BS finetuning method applied to a BS-trained backbone can still result in significant improvements, indicating the importance of second stage training. **As a plug-and-play adjustment method, our approach is capable of effectively improving the performance of other models by finetuning the classifier.**

## 5 CONCLUSION

In this paper, we conduct a thorough analysis of current classifier retraining methodologies and introduce two innovative metrics: "Logits Magnitude" and "Regularized Standard Deviation." These metrics offer fresh insights into model performance and highlight the key requirements for achieving enhanced accuracy. Consequently, we propose a "Label Over-Smooth" (LOS), which systematically adjusts the optimization targets for logits and achieves more balanced Logits Magnitude. Our extensive experimental evaluation demonstrates that LOS achieves state-of-the-art performance on various long-tailed recognition datasets. At the same time, LOS not only adapts to our theoretical analysis but also serves as a regularization constraint on the model's capacity. Therefore, there may be opportunities in the future to explore more suitable methods.

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

## A    RELATED WORK

**Long-Tailed Recognition (LTR).** In real-world scenarios, the collected data exhibits a long-tail distribution, wherein a small fraction of classes possess a substantial number of samples and the remaining classes are associated with only a few samples (Zhang et al., 2023). To address long-tailed class imbalance and the failure of the vanilla empirical risk minimization methods, massive studies have been conducted recently. (i) *Re-sampling* aims to balance the training samples of different class (Kang et al., 2019; Wang et al., 2020a; Mahajan et al., 2018; Ren et al., 2020) by over-sampling the data from the tail class and under-sampling the data from the head class. (ii) *Cost-sensitive learning* adjusts loss values for different classes to make a balance, such as focal loss (Lin et al., 2017), class-balanced loss (CB) (Cui et al., 2019), balanced softmax loss (BS) (Ren et al., 2020), LADE (Hong et al., 2021), LDAM (Cao et al., 2019) and logit adjustment loss (Zhao et al., 2022; Li et al., 2022b). (iii) *Representation Learning* focuses on improving the feature extractors, consisting some paradigms, i.e., metric learning (Huang et al., 2016; Zhang et al., 2017), prototype learning (Liu et al., 2019; Zhu & Yang, 2020), and transfer learning (Zhong et al., 2021; Li et al., 2021). (iv) *Classifier Design* replaces the linear classifier with well-designed classifiers, such as scale-invariant cosine classifier (Wu et al., 2021) and $\tau$-normalized classifier (Kang et al., 2019). (v) *Decoupled Training* decouples the learning into representation learning and classifier retraining, rather than training them jointly. (vi) Other methods use data augmentation (Zada et al., 2022; Zhong et al., 2021; Perez, 2017; Shorten & Khoshgoftaar, 2019; Chou et al., 2020), (Li et al., 2022a; Wang et al., 2024), self-supervised learning (Liu et al., 2021; Cui et al., 2021), (Suh & Seo, 2023), (Du et al., 2024), multi-expert ensemble (Cai et al., 2021; Cui et al., 2022; Zhang et al., 2021b), pretrained models (Ru et al., 2024; Shi et al., 2024) and so on (Zhang et al., 2023). These models require more data or computation resources and are inclined to achieve high accuracy. In the paper, we report them as the upper bound baseline.

**Decoupled Training.** The Decoupled Training scheme (Alshammari et al., 2022; Cui et al., 2019; Zhang et al., 2021a; Ren et al., 2020) in LTR separates the joint learning process into feature learning and classifier learning stages. Decoupling (Kang et al., 2019) is the pioneering work to experimentally evaluate different sampling strategies in the first stage and classifier design (e.g. cRT and $\tau$-norm) in the second stage. KCL (Kang et al., 2021) empirically finds a balanced feature space is important for LTR. MiSLAS (Zhong et al., 2021) observes that data mixup benefits the feature extractor in the first stage but harms classifier performance in the second stage. DisAlign (Zhang et al., 2021a) develops the distribution alignment strategy with a generalized re-weight method. (LTWB) (Alshammari et al., 2022) emphasizes parameter regularization in both feature learning and classifier finetuning, achieving state-of-the-art performance.

## B    PROOF OF PROPOSITION 1 BIAS CONVERGENCE

*Proof.* The loss function is

$$\mathcal{L} = \sum_i -\mathbf{y}_i \cdot \log\left(\frac{e^{\mathbf{z}_i}}{\sum_j e^{\mathbf{z}_j}}\right) \tag{13}$$

Using the Chain Rule to calculate derivatives,

$$
\begin{aligned}
\frac{\partial \mathcal{L}}{\partial \mathbf{z}_i} &= \sum_j \frac{\partial \mathcal{L}}{\partial \mathbf{s}_j}\frac{\partial \mathbf{s}_j}{\partial \mathbf{z}_i} \\
&= \frac{\partial \mathcal{L}}{\partial \mathbf{s}_i}\frac{\partial \mathbf{s}_i}{\partial \mathbf{z}_i} + \sum_{j\neq i} \frac{\partial \mathcal{L}}{\partial \mathbf{s}_j}\frac{\partial \mathbf{s}_j}{\partial \mathbf{z}_i} \\
&= -\frac{\mathbf{y}_i}{\mathbf{s}_i}(\mathbf{s}_i(1-\mathbf{s}_i)) - \sum_{j\neq i}\frac{\mathbf{y}_j}{\mathbf{s}_j}(-\mathbf{s}_i\mathbf{s}_j) \\
&= \mathbf{y}_i(\mathbf{s}_i - 1) + \sum_{j\neq i}\mathbf{y}_j\mathbf{s}_i \\
&= \mathbf{s}_i - \mathbf{y}_i
\end{aligned}
\tag{14}
$$

Therefore, the Hessian matrix of $\mathcal{L}$ with respect to $\mathbf{b}$ as $\mathbf{H}$ is as follows:

$$\mathbf{H}_{ij} = \begin{cases} \mathbf{s}_i(1 - \mathbf{s}_i) & \text{if } i = j \\ -\mathbf{s}_i\mathbf{s}_j & \text{if } i \neq j \end{cases} \tag{15}$$

And for any vector $\mathbf{x}$,

$$\mathbf{x}\mathbf{H}\mathbf{x}^\top = \sum_i \mathbf{s}_i\mathbf{x}_i^2 - \left(\sum_i \mathbf{s}_i\mathbf{x}_i\right)^2 \tag{16}$$

Since $g(\mathbf{x}) = \mathbf{x}^2$ is a convex function, with $\mathbf{s}_i > 0$ and $\sum \mathbf{s}_i = 1$, by Jensen's Inequality:

$$\sum_i \mathbf{s}_i\mathbf{x}_i^2 \geqslant \left(\sum_i \mathbf{s}_i\mathbf{x}_i\right)^2, \tag{17}$$
$$\mathbf{x}\mathbf{H}\mathbf{x}^\top \geqslant 0$$

Therefore, the Hessian matrix $\mathbf{H}$ is positive semi-definite matrix. $\square$

## C  ANALYZING THE APPROXIMATION IN EQ. 9

The fomulation:

$$\mathbb{E}\left[\frac{1}{\sum_j e^{(\Delta_j - \Delta_i)}e^{(z_j - z_i)}}\right] \approx \frac{1}{\mathbb{E}[e^{\Delta_j - \Delta_i}]}\mathbb{E}\left[\frac{1}{\sum_j e^{z_j - z_i}}\right] \tag{18}$$

For convenience, let $p_j = \Delta_i - \Delta_j$, $q_j = z_j - z_i$,

$$\mathbb{E}\left[\frac{1}{\sum_j e^{p_j}e^{q_j}}\right] \approx \frac{1}{\mathbb{E}[e^{p_j}]}\mathbb{E}\left[\frac{1}{\sum_j e^{q_j}}\right] \tag{19}$$

$$\mathbb{E}\left[\frac{1}{\sum_j e^{p_j}e^{q_j}}\right] = \mathbb{E}\left[\frac{1}{\mathbb{E}[e^{p_j}]\sum_j e^{q_j} + \sum_j(e^{p_j} - \mathbb{E}[e^{p_j}])e^{q_j}}\right] \tag{20}$$

Assuming that the fluctuations $e^{p_j} - \mathbb{E}[e^{p_j}]$ are small compared to $\mathbb{E}[e^{p_j}]$, we can use the first-order approximation of the reciprocal:

$$\frac{1}{A + \delta} = \frac{1}{A} - \frac{\delta}{A^2} + \frac{\delta^2}{A^3} + \dots \tag{21}$$
$$\approx \frac{1}{A} - \frac{\delta}{A^2} \tag{22}$$

$$\mathbb{E}\left[\frac{1}{\sum_j e^{p_j}e^{q_j}}\right] \approx \mathbb{E}\left[\frac{1}{\mathbb{E}[e^{p_j}]\sum_j e^{q_j}}\right] - \mathbb{E}\left[\frac{\sum_j(e^{p_j} - \mathbb{E}[e^{p_j}])e^{q_j}}{(\mathbb{E}[e^{p_j}]\sum_j e^{q_j})^2}\right] \tag{23}$$

Since $\mathbb{E}[e^{p_j} - \mathbb{E}[e^{p_j}]] = 0$,

$$\mathbb{E}\left[\frac{1}{\sum_j e^{p_j}e^{q_j}}\right] \approx \mathbb{E}\left[\frac{1}{\mathbb{E}[e^{p_j}]\sum_j e^{q_j}}\right] \tag{24}$$

## D   DATASETS AND IMPLEMENTATION DETAILS

We define $K$ as class number and imbalanced ratio (IR) of the long-tailed datasets as IR $= n_{max}/n_{min}$, where $n_{max}$ and $n_{min}$ is the number of training samples in the largest and smallest class, respectively.

**Datasets.** 1) CIFAR100-LT: The original balanced CIFAR100 (Krizhevsky et al., 2009) consists of 50,000 training images and 10,000 test images of size 32×32 with 100 classes. Following (Cao et al., 2019), the long-tailed version CIFAR100-LT was constructed by exponentially decaying sampling for the training images of each class, maintaining the balanced test set unchanged simultaneously. Setting the imbalanced factor IR to 10/50/100, we create three long-tailed training datasets as (Alshammari et al., 2022). 2) ImageNet-LT is built in (Liu et al., 2019) from ImageNet dataset (Russakovsky et al., 2015) with 1000 classes. The number of the images for each class ranges from 5 to 1280, which means the imbalanced factor IR=256. 3) iNaturalist2018 (Van Horn et al., 2018) is a real-world, large-scale dataset for species identification of animals and plants with 438K images for 8142 classes. The imbalanced factor is extremely large as 500.

**Implementation.** In line with the SOTA method (Alshammari et al., 2022), we use ResNet34 (He et al., 2016) in CIFAR100-LT, ResNeXt50 (Xie et al., 2017) in ImageNet-LT and ResNet50 (He et al., 2016) in iNaturalist2018. For each model, we use SGD optimizer with momentum 0.9 and cosine learning rate scheduler (Loshchilov & Hutter, 2016) from given initial learning rate to zero. We use Decoupled Training Scheme which consists of training the feature extractor and finetune the classifier separately. The feature extractor is trained with cross-entropy (CE) loss and suitable weight decay follows (Alshammari et al., 2022) and the classifier is finetuned using our proposed method. On CIFAR100-LT, We train each model for 200 epochs, with batch size 64 and initial learning rate 0.01. On ImageNet-LT / iNaturalist2018, we train for 200 epochs, with batch size as 128 / 256 and initial learning rate 0.03 / 0.1. We use random left-right flipping and cropping as our training augmentation, left-right flipping as testing augmentation for the three datasets with the same configuration as (Alshammari et al., 2022). In ImageNet-LT, we also use the color jitter following (Cui et al., 2022). For smooth factors, we use 0.98 in CIFAR100-LT and 0.99 in ImageNet-LT and iNaturalist2018. The classifier finetune epoch is 20 with adequate learning ratio and weight. We report the best result in five trails. Results in Tab. 1 for CIFAR100LT with IR=100 also follow this implementation.

## E   SENSITIVITY OF LEARNING RATE AND WEIGHT DECAY

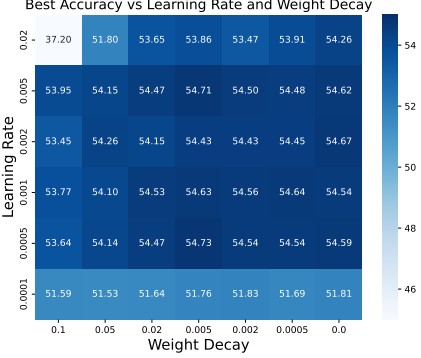

Figure 5: **Sensitivity Comparison of Learning Rate and Weight Decay**.

The adjustable hyper-parameters in classifier finetuning are learning rate and weight decay. We use grid search to find the best accuracy and report the sensitivity using heat map. According to Fig. 5 with the smooth factor as 0.99, it is evident that our method is robust and can achieve a result close to the optimal solution under different combinations. **Therefore, our proposed model exhibits stability and ease of convergence.**

