# OpenReview forum: "Rethinking Classifier Re-Training in Long-Tailed Recognition: Label Over-Smooth Can Balance"
_ICLR.cc/2025/Conference — ICLR 2025 Poster_

### Official Review · Reviewer_o1sQ · 2024-11-02

**Soundness:** 3
**Presentation:** 3
**Contribution:** 3
**Rating:** 8
**Confidence:** 3

**Summary:**

This paper revisits the classifier re-training in long-tailed recognition, proposing a simple label over-smoothing approach to balance the logits magnitude across classes. The method achieves state-of-the-art performance on various imbalanced datasets.

**Strengths:**

1. The method is simple, using basic label smoothing to effectively reduce overfitting.
1. Theoretical analysis and experimental validation are comprehensive, demonstrating the method's effectiveness across various datasets.

**Weaknesses:**

There are no significant methodological drawbacks. It is recommended that the authors include comparisons with the latest SOTA methods^[1] and discuss related works^[2,3].

[1] Probabilistic Contrastive Learning for Long-Tailed Visual Recognition. TPAMI 2024.

[2] Adaptive Logit Adjustment Loss for Long-Tailed Visual Recognition. AAAI 2022.

[3] Long-Tailed Visual Recognition via Gaussian Clouded Logit Adjustment. CVPR 2022.

**Questions:**

See the weaknesses section.

---

> ### Author Response · Authors · 2024-11-19
> **Response to Reviewer o1sQ**
>
> >***Q1:  It is recommended that the authors include comparisons with the latest SOTA methods and discuss related works.***
>
> **R1:** Thank you very much for recognizing the value of our work! We also appreciate the suggested references that help us further improve our paper. [2][3] represent classical works in this field and the newer work [1] certainly deserves inclusion. Let us first discuss our relationship with [2][3], and then incorporate the SOTA results from [1] into our performance comparisons. **We will add these works to the paper in the revison.**
>
> Adaptive Logit Adjustment Loss (ALA Loss)[2] **innovatively applies an adaptive adjusting term to the logit**. The adaptive adjusting term comprises two complementary factors: 1) a quantity factor that emphasizes tail classes, and 2) a difficulty factor that adaptively focuses on hard instances during training. This approach integrates data priors and Hard Data Mining concepts. In contrast, **our method maintains purity and simplicity**. We focus more on optimizing the overall class distribution without relying on artificial priors, demonstrating superior versatility and effectiveness. Moreover, **our approach is orthogonal to the Difficulty Factor design mentioned in [2], suggesting potential for integration**.
>
> [3] begins **from the perspective of feature space**, aiming to expand the coverage of minority classes within this space. [3] improves feature calibration through logits. Their optimization involves **shifting the predicted logits values** $z$ based on class nums. From a motivation standpoint, we propose **a more effective metric directly derived from the logits**, around which we develop our work by optimizing this specific metric. In terms of optimization techniques, unlike the shift of $z$ during training in [3], our method actually implements label over-smooth, which **introduces new predictive targets** to replace the traditional one-hot ground truth vectors.
>
> Our method and analysis focus on the properties and disparities in the training process, particularly the potential convergence behaviors induced by head and tail classes. Indeed, a key advantage of our method is **its independence from class-specific hyperparameters** to mitigate long-tail issues rather than the parameters which is handcrafted in the final equations of [2][3].
>
>
>
> | Method                      | Reference | CIFAR100-LT |            |            | ImageNet-LT |
> | --------------------------- | --------- | ----------- | ---------- | ---------- | ---------- |
> |                             |           | IR=100      | IR=50      | IR=10   |        |
> | **LOS (ours)** |           | **54.9**    | **58.8**   | **69.7**  | 54.4   |
> | ProCo           |   TPAMI24        | 52.8  | 57.1 | 65.5  |  58.0       |
> | **ProCo+LOS(ours)**                      |        | 53.9(+1.1)  | 58.0(+0.9) | 66.7(+1.2)  | **58.5(+0.5)**       |
>
> This table demonstrates the comparison between our method and ProCo[1]. ProCo[1] employs contrastive learning and requires AutoAugment, **utilizing substantially higher computational costs and data augmentation compared to our approach.** We position it as an Upperbound Model, similar to those in Tab.3 of our main paper.
>
> Consequently, **ProCo[1] can learn better feature representations**, giving it an advantage on challenging datasets like ImageNet-LT. However, **its classifier performance limitations become evident in CIFAR100-LT**. Without employing any additional tricks, **our method significantly outperforms [1]. Furthermore, leveraging ProCo's excellent feature representations, we can integrate our method as a plugin to adjust its classifier, achieving notable performance improvements across different tasks**. This demonstrates the complementary nature of our approach and its ability to enhance even sophisticated methods through better classifier optimization.
>
>
>
> [1] Probabilistic Contrastive Learning for Long-Tailed Visual Recognition. TPAMI 2024.
>
> [2] Adaptive Logit Adjustment Loss for Long-Tailed Visual Recognition. AAAI 2022.
>
> [3] Long-Tailed Visual Recognition via Gaussian Clouded Logit Adjustment. CVPR 2022.

---

### Official Review · Reviewer_aP5D · 2024-11-03

**Soundness:** 3
**Presentation:** 2
**Contribution:** 3
**Rating:** 6
**Confidence:** 4

**Summary:**

This paper proposes a simple but effective method for long-tailed problem. They base their method in decoupled ones and first propose two new metrics, Logits Magnitude and Regularized Standard Deviation, to compare the differences and similarities between various methods. Then, they propose a simple method by softening the original one-hot labels by assigning a probability slightly higher than $1/K$ to the true class and slightly lower than $1/K$ to the other classes, where $K$ is the number of classes. In the method, they also provide the detailed analysis how the method is designed. In the experiments, the results validate the effectiveness of proposed method.

**Strengths:**

(1) The structure of this paper is clear, starting from the definition of problem, the background motivation and analysis to the final implementation.

(2) The experiments are good. By applying the proposed method to some existing methods, their performances can be further improved.

**Weaknesses:**

Although the structure is clear, the transition from last part to the next can be further improved.

Some of the explanations are also missed. See below.

**Questions:**

I tried to understand the logic and the details of this paper, so some of the points in the paper require further clarification:

(1) In Proposition 1. It is said " In all cross-entropy based methods, the aforementioned conclusion remains valid", in deep learning networks, the objective is always a non-convex optimization problem, how the conclusion is still valid in this case?

(2) The Proposition 1 is used to exclude the effect of b and to introduce Proposition 2 which illustrates that the weight norm-based method may not be reliable in some cases, am I right? But how the equation (4) is obtained? from s' to s.

(3) From line 313-314, how the assumption is determined" Consider random perturbations ∆i that independently affect each class zi and are proportional to the standard deviation of their logits, i.e., z′"?? It is said "These perturbations ∆ are commonly introduced during the training process, particularly due to factors such as overfitting in the initial stage of training and bias in the sampling of the training set.", can you clarify more on this point?

(4) if the point lies in $\Delta$ and $\sigma$ in section 3.1, what is the point to introduce definition 2?? The point in definition 2 shoule be $R_i$, right?

(5) Figure 1 is also not easy to understand for the first time until figure 2 is given.

Typo: (1) Is the parenthesis missed in line 107?

---

> ### Author Response · Authors · 2024-11-19
> **Response to Reviewer aP5D**
>
> >***Q1: In deep learning networks, the objective is always a non-convex optimization problem, how the conclusion is still valid in this case in Proposition 1?***
>
> **R1:** In proposition 1, we assumed that $W$ is given and determined and $b$ is the only variable. With such assumption, the optimization problem is convex with respect to $b$ as we proved. This shows that we don't need to pay too much attention to $b$.
>
> >***Q2:  The Proposition 1 is used to exclude the effect of b and to introduce Proposition 2 which illustrates that the weight norm-based method may not be reliable in some cases, am I right? But how the equation (4) is obtained? from s' to s.***
>
> **R2:** Right, Proposition 2 illustrates that the weight norm-based method may not be reliable and compare the weight across classes is unreliable. That is the reason that we proposed LoMa. In Proposition 2, We just construct a new converage point $(W', b^*)$. In such situation, the calculated new $s'$ is not changed from original $s$. By substituting into equation (2), we can get equation (4).
>
> >***Q3:From line 313-314, how the assumption is determined" Consider random perturbations ∆i that independently affect each class zi and are proportional to the standard deviation of their logits, i.e., z′"? It is said "These perturbations ∆ are commonly introduced during the training process, particularly due to factors such as overfitting in the initial stage of training and bias in the sampling of the training set.", can you clarify more on this point?***
>
> **R3:**
>
> Consider the basic derivative
> $$
> \frac{\partial \mathcal{L}}{\partial z_i}= (s_i-y_i)
> $$
> $$
> \frac{\partial \mathcal{L}}{\partial w_i}= (s_i-y_i)f_i
> $$
> $$
> \frac{\partial \mathcal{L}}{\partial b_i}= (s_i-y_i)
> $$
>
> They both converage as $\mathbb{E}[s_i] = \mathbb{E}[y_i]$,
>
> In each step,
> $$
> \mathbb{E}_w[w_i'] = \mathbb{E}_w[w_i] - \text{lr} \cdot \mathbb{E}_w[(s_i-y_i)f_i]
> $$
> $$
> \mathbb{E}_w[w_i'] = \mathbb{E}_w[w_i] - \text{lr} \cdot \mathbb{E}_w[s_i]f_i + \text{lr} \cdot \mathbb{E}_w[y_i]f_i
> $$
>
> $\mathbb{E}_w[s_i]$ is widely present during the gradient descent process, so its inconsistent magnitude will lead to differences in convergence speeds in different directions and an overall directional bias relative to balanced datasets.
>
> Due to the randomness of the sampling itself, or the bias of $W$ after first stage of training, even the SGD optimizer will introduce such perturbations.
>
> We will add more explanation in the revision to help understanding.
>
> >***Q4: If the point lies in $\Delta$ and $\sigma$ in section 3.1, what is the point to introduce definition 2? The point in definition 2 shoule be $R_i$, right?***
>
> **R4:** With definition 2, we observe $R_i$ is almost invariant between these methods as $\sigma$ is not. Based on the invariance of $\sigma$ and the balance of LoMa $L$, we can analyze the trend of $\sigma$ in following.
>
>
> >***Q5: Figure 1 is also not easy to understand for the first time until figure 2 is given.***
>
> **R5:** Thank you for pointing out this issue. We will add more explanation in figure 1 in the revision to help understanding.
>
> >***Q6: Typos Line 107.***
>
> **R6:** Thank you for pointing out this issue. We mistakenly used `\cite` when it should have been `\citep`. We have made this correction in the revised PDF version.

---

> > ### Comment · Reviewer_aP5D · 2024-11-25
> > **Thank you for response.**
> >
> > Thank you for the responses.
> >
> > (1) For Q2, will it satisfy naturally? Any proof?
> > (2) For Q3, what do you mean " $R_i$ is almost invariant between these methods as $\sigma$ is not"? $\sigma$ is not invariant??
> > (3) It is suggested to provide the line number about the changes in the paper. It would be easier to track what and where have been revised?

---

> ### Author Response · Authors · 2024-11-25
>
> >***Q1: For Q2, will it satisfy naturally? Any proof?***
>
> **R1:** For Proposition 1, the proof is given in the Appendix B. Equation 4 is part of the proof for Propostion 2, the new converage points $(W', b^*)$ are constructed to illustrate the existence. Such converage points naturally exist. You may notice the definetion of $z$ in Section 2.1, from line 146 to 153, Equation 4 is obtained from the given definition. I may not understand what "it" refers to, I hope this solves your doubts.
>
> >***Q2: For Q3, what do you mean "$R_i$ is almost invariant between these methods as $\sigma$ is not"? $\sigma$ is not invariant?***
>
> **R2:** Since $R_i$ is defined as
>
> $$ R_i=\frac{\sigma_i}{L_i},$$
>
> where $L_i$ is logits magnitude. $R_i$ is invariant and $L_i$ is variant across methods as shown in Figure 3. So $\sigma_i=L_iR_i$ is not invariant as well.
>
> >***Q3: It is suggested to provide the line number about the changes in the paper. It would be easier to track what and where have been revised?***
>
> **R3:** In the uploaded revision PDF, **all modifications have been highlighted in blue.** Below are the specific line numbers and corresponding changes:
>
> 1. **Line80-87 & Figure1:** Replace "Gaussian distribution diagram drawn with mean and variance of samples" with a "bar chart showing specific logits value for each data samples". Update the caption to reflect these changes.
>
> 2. **Line168:** Generalize Vanilla Cosine to a more general form, add temperature parameter $\tau$, conducte new experiments and update best results for this method.
>
> 3. **Line325 & Appendix C:** Add proof for the approximation in Equation 9.
>
> 4. **Line215 & Line307:** Modify expressions for improved readability.
>
> 5. **Line440-445 (Table3):** Add new experimental results combining the latest methods with our approach.
>
> 6. **Line457-Line459, Line766, Line773-776:** Add analysis of latest related work.
>
> 7. **Line540-Line715 (Reference):** Standardize paper citation format, update journal/conference names and abbreviations and replace arXiv versions with published versions.
>
> *Note: Figure sizes have been adjusted to maintain the 10-page limit without affecting crucial content.*

---

> ### Comment · Reviewer_aP5D · 2024-11-28
> **Thank you for response.**
>
> I think the authors have addressed my concerns. I can raise the score to 6.
>
> One last thing, please pay attention to R1 "Equation 4 is part of proof in for Propostion 2", the typo.

---

> ### Author Response · Authors · 2024-11-28
>
> Thank you very much for your time and constructive feedback, which helped us effectively improve this paper. We truly appreciate your recognition and positive evaluation of our work.
>
> P.S. We have modified the typo as suggested - thank you for pointing that out.

---

### Official Review · Reviewer_g4ig · 2024-11-04

**Soundness:** 3
**Presentation:** 3
**Contribution:** 3
**Rating:** 5
**Confidence:** 5

**Summary:**

This paper revisits the paradigm of classifier re-training in the context of long-tailed recognition and proposes two metrics for analyzing the efficacy of classifiers: logits magnitude (LoMa) and regularized standard deviation. Various existing methods for LTR are re-evaluated with a unified feature representation and with respect to the proposed metrics. They further use a variant of label smoothing where very strong smoothing is used to achieve better performance. Results and ablation studies are reported on multiple benchmark LTR datasets.

**Strengths:**

1. The performance of various methods seems to somewhat correlate to the proposed metrics.
2. The proposed method outperforms prior arts on multiple benchmarks in most of the evaluated settings.
3. The over-smoothing technique can be used as a plugin on top of existing methods.

**Weaknesses:**

1. The proposed LoMa metric is hard to understand due to ambiguous language and notation. The authors should use two indices in the subscript of the logits to differentiate between sample and class. In Fig. 2 it is unclear what true samples and non-true samples are.
2. While the authors claim it is hard to optimize directly for LoMa or Regularized Standard Deviation, they don't propose any way to do this. The suggested label smoothing approach doesn't directly fit into the story of achieving uniform LoMa.
3. There are wild claims made, for instance, that classifier weight norm cannot capture a method's efficacy due to potential arbitrary lengths (Proposition 2). It is relative weight norm that matters, not absolute, thus rendering Prop. 2 meaningless. Section 3.1 on the deep dive into LoMa seems like a rambling section without concrete takeaways.
4. The math in the paper could be improved. For instance, in Eq. 9, the expectation can't be decomposed like that, it looks artificial.
5. The proposed (over) label smoothing technique is just a minor variant of prior work.

**Questions:**

Please address the points in weaknesses above.

---

> ### Author Response · Authors · 2024-11-19
> **Response to Reviewer g4ig**
>
> >***Q1: The proposed LoMa metric is hard to understand due to ambiguous language and notation.***
>
> **R1:**
> Each Loma $L_i \in R$ is uniquely associated with a single class $c_i$. As shown in Figure 2, the Loma $L_i$ is defined as the difference between the mean logits of the samples belonging to the current class $c_i$ and and those of the samples not belonging to $c_i$.
>
> >***Q2: The suggested label smoothing approach doesn't directly fit into the story of achieving uniform LoMa.***
>
> **R2:**
>
> 1) We conducted an analysis of previous methods and concluded the following: **A balanced LoMa typically leads to better performance. The Regularized Standard Deviation (RSD) is invariant**, meaning it does not significantly vary across different methods(Section 2.2). Moreover, it remains balanced on datasets that are themselves balanced.
> 2）Building on these observations, we analyzed the differences in the gradient descent process **between balanced datasets and long-tailed datasets (Section 3.1).**
> 3) As a non-convex optimization problem, gradient descent inevitably deviates from the true optimal point and direction. **On balanced datasets, these deviations can be considered consistent across classes (Eq. 2). However, on long-tailed datasets, noticeable inter-class inconsistencies emerge.**
> 4) To address this issue, we propose a method that not only maintains a well-balanced LoMa but also eliminates inter-class inconsistencies.
>
> >***Q3: Proposition 2 is meaningless. Section 3.1 on the deep dive into LoMa seems like a rambling section without concrete takeaways.***
>
> **R3:** Proposition 2 demonstrates that the convergence point is not unique, and the norm $||w_i||$ can have arbitrary length at convergence. Many methods attempt to constrain the length of $||w||$. However, based on Proposition 2，**directly comparing $||w_i||^2_2$ across classes is meaningless.** To address this, we propose LoMa，which enables unified comparison across classes. It is worth noting that LoMa is also related to $w$. For instance, simply doubling $w_i$ would result in the corresponding LoMa $L_i$ being doubled.
>
> >***Q4: In Eq. 9, the expectation can't be decomposed like that, it looks artificial.***
>
> **R4:** In Eq. 9, we assume a balanced dataset scenario, meaning that $\Delta_i$ and $\Delta_j$ are identically distributed.
>
> **Analyzing the Approximation:**
>
> $$
> \mathbb{E}\left[\frac{1}{\sum_j e^{(\Delta_j - \Delta_i)} e^{(z_j - z_i)}}\right] \approx \frac{1}{\mathbb{E}[e^{\Delta_j - \Delta_i}]}  \mathbb{E}\left[\frac{1}{\sum_j e^{z_j - z_i}}\right]
> $$
>
> For convenience, let $p_j = \Delta_i-\Delta_j$， $q_j=z_j-z_i$,
> $$
> \mathbb{E}\left[\frac{1}{\sum_j e^{p_j} e^{q_j}}\right] \approx \frac{1}{\mathbb{E}[e^{p_j}]} \mathbb{E}\left[\frac{1}{\sum_j e^{q_j}}\right]
> $$
>
>
> \begin{align}
> \mathbb{E}\left[\frac{1}{\sum_j e^{p_j} e^{q_j}}\right]&= \mathbb{E}\left[\frac{1}{\mathbb{E}[e^{p_j}]\sum_j e^{q_j} + \sum_j (e^{p_j} - \mathbb{E}[e^{p_j}])e^{q_j}}\right]
> \end{align}
>
>
> Assuming that the fluctuations $e^{p_j} - \mathbb{E}[e^{p_j}]$ are small compared to $\mathbb{E}[e^{p_j}]$, we can use the first-order approximation of the reciprocal:
>
> \begin{align}
> \frac{1}{A+\delta} &= \frac{1}{A} - \frac{\delta}{A^2} + \frac{\delta^2}{A^3} + \dots    \\\\
> &\approx \frac{1}{A} - \frac{\delta}{A^2}
> \end{align}
>
> \begin{align}
> \mathbb{E}\left[\frac{1}{\sum_j e^{p_j} e^{q_j}}\right]&\approx \mathbb{E}\left[\frac{1}{\mathbb{E}[e^{p_j}]\sum_j e^{q_j}}\right] - \mathbb{E}\left[\frac{\sum_j (e^{p_j} - \mathbb{E}[e^{p_j}])e^{q_j}}{(\mathbb{E}[e^{p_j}]\sum_je^{q_j})^2}\right] \\\\
> \end{align}
>
> Since $\mathbb{E}[e^{p_j} - \mathbb{E}[e^{p_j}]]=0$,
>
> \begin{align}
> \mathbb{E}\left[\frac{1}{\sum_j e^{p_j} e^{q_j}}\right]&\approx \mathbb{E}\left[\frac{1}{\mathbb{E}[e^{p_j}]\sum_j e^{q_j}}\right]
> \end{align}
>
> >***Q5: The proposed (over) label smoothing technique is just a minor variant of prior work.***
>
> **R5:** Although our proposed method is a variant of prior work, its usage is completely different. We not only proposed the concept of over-label smoothing but also theoretically analyzed its impact on the model learning process. **Label smoothing has not demonstrated a dominant position in the long-tailed (LT) field, and no one has attempted over-smoothing in other fields.** The difference between 0.99 and 0.2 has already far exceeded mere formal consistency. Our method is simple but effective. It can be extremely effective in the LT domain, which constitutes a novelty.

---

> > ### Author Response · Authors · 2024-11-26
> >
> > Thank you for your insightful suggestions. We believe we have comprehensively addressed your questions regarding the definition of LoMa Metric, the motivation and novelty of the method design, the significance of Proposition 2, and the proof of Eq.9.
> >
> > We are wondering whether you have any additional questions or comments regarding our response to your review comments. We will do our best to address them.
> >
> > We appreciate your time and effort reviewing our manuscript. Thanks for your consideration!

---

> ### Author Response · Authors · 2024-11-30
> **Follow-up on Rebuttal Discussion**
>
> I hope this message finds you well. I am following up on my previous response to your review comments on our manuscript. We deeply value your feedback and aim to ensure we address all your concerns thoroughly.
>
> If there are any further questions or clarifications you would like us to address, we are more than happy to provide additional details.

---

### Official Review · Reviewer_sTG2 · 2024-11-05

**Soundness:** 3
**Presentation:** 3
**Contribution:** 3
**Rating:** 6
**Confidence:** 4

**Summary:**

This paper revisits the two-stage learning paradigm in long-tailed recognition and proposes two metrics, Logits Magnitude, and Regularized Standard Deviation, to compare different classifier re-training methods. Moreover, it develops a label over-smoothing approach to improve model performance. Experimental results on multiple long-tailed datasets demonstrate the effectiveness of the proposed method.

**Strengths:**

- This paper reveals some shortages in previous metrics, such as the ignoration of feature magnitude.

- The proposed label over-smoothing is interesting. Both theoretical and empirical results demonstrate the effectiveness.

- The proposed LOS outperforms most compared methods on multiple datasets. The ablation studies are well done.

**Weaknesses:**

- Figure 1 is hard to follow. What do the red and blue lines mean? Why do they obey Gaussian distributions? More explanations are required.

- Since the feature magnitude and the class weight norm both have effects on prediction results, why not directly use the Cosine classifier? The Cosine classifier is quite simple and effective, which calculates Cosine similarities between the normalized features and the normalized class weights, and then divides it by a temperature to obtain the final logits. Therefore, it can directly mitigate the impact of feature magnitude. (Well, maybe the logit magnitude is different from the feature magnitude, but I suggest the authors include the Cosine classifier as a baseline.)

- Despite the superior performance. Some recent works are ignored and not compared. For example, RIDE, BCL, NCL, GML.

- The references are outdated. Among the 50 cited papers, there are no papers from 2024, only 3 papers from 2023, and 4 papers from 2022.

- Please pay attention to the format of the references. Some do not include the source (e.g. Peifeng et al., 2023); some have a URL (Huang et al., 2018) but others do not; some source names are inconsistent (e.g. "Computer Vision-ECCV ..." or "Proceedings of ... (ECCV)"? and "arXiv preprint" or "ArXiv, abs/..." or "arXiv e-prints"?)

**Questions:**

See "Weaknesses".

---

> ### Author Response · Authors · 2024-11-19
> **Response to Reviewer sTG2 [1/2]**
>
> >***Q1: In Fig1, what do the red and blue lines mean? Why do they obey Gaussian distributions?***
>
> **R1:** The red line shows the distribution of samples that truly belong to the current class, while the blue line shows the distribution of samples that don’t.
>
> It cannot be proven that they follow a Gaussian distribution, but their actual distribution is very **similar to a Gaussian distribution**. Here, **as a qualitative demonstration**, we directly used the mean and variance to plot a Gaussian distribution.
>
> >***Q2: Why not directly use the Cosine classifier? Should include the Cosine classifier as a baseline.***
>
> **R2:** Thank you for your sincere suggestion! In fact, **we have already included the Cosine classifier as a baseline for comparison in Tab.1 of the main paper**, and analyzed its characteristics in the subsequent Fig.3. Here we rewrite the experimental results.
>
> | Methods        | All   | Many  | Medium | Few   |
> | -------------- | ----- | ----- | ------ | ----- |
> | Vanilla FC     | 48.44 | 78.74 | 47.37  | 14.33 |
> | Vanilla Cosine | 51.66 | 77.77 | 49.60  | 23.60 |
> | Ours | 54.94 | 74.94  | 52.91 | 33.96    |
>
> As shown in the table, our method significantly outperforms the Cosine method. Indeed, as you mentioned, feature magnitude and logit magnitude are not the same. We focus more on the relationships between computed Logits, directly optimizing a more straightforward objective, rather than pre-normalizing Weights and Features.
>
>
> >***Q3: Comparison to recent works RIDE, BCL, NCL, GML.***
>
> **R3:** **In the main paper Tab.3**, we compared with these Upperbound models, which typically require larger models (e.g. Mixture of Experts) and more computational resources (e.g. Contrastive Learning). This **includes models like RIDE, BCL**, PaCo, and SSD.
>
> Following your suggestion, we have supplemented the **complete results for NCL and GML**, and presented the unified results below.
>
> | Method                      | Reference | CIFAR100-LT |            |            | ImageNet-LT |            |            |            |
> | --------------------------- | --------- | ----------- | ---------- | ---------- | ----------- | ---------- | ---------- | ---------- |
> |                             |           | IR=100      | IR=50      | IR=10      | Many        | Medium     | Few        | All        |
> | **Vallina CE + LOS (ours)** |           | **54.9**    | **58.8**   | **69.7**   | **63.2**    | **50.7**   | **42.3**   | **54.4**   |
> | RIDE                        | ICLR20    | 49.1        | ---        | ---        | 67.9        | 52.3       | 36.0       | 56.1       |
> | BCL                         | CVPR22    | 51.0        | 54.9       | 64.4       | 65.3        | 53.5       | 36.3       | 55.6       |
> | NCL[1]                      | CVPR22    | 53.3        | 56.8       | ---        | ---         | ---        | ---        | 57.4       |
> | GML[2]                      | ICML23    | 53.0        | 57.6       | 65.7       | 68.7        | 55.7       | 38.6       | 58.3       |
> | **RIDE+LOS(ours)**          |           | 50.2(+1.1)  | ---        | ---        | 66.9(-1.0)  | 53.2(+0.9) | 37.9(+1.9) | 56.5(+0.4) |
> | **BCL+LOS(ours)**           |           | 52.2(+1.2)  | 56.0(+1.1) | 65.1(+0.7) | 64.7(-0.6)  | 54.1(+0.8) | 28.1(+1.8) | 56.1(+0.5) |
> | **NCL+LOS(ours)**                      |     | 54.1(+0.8)        | 57.5(+0.7)       | ---        | ---         | ---        | ---        | 57.7(+0.3)       |
> | **GML+LOS(ours)**                      |     | 54.0(+1.0)        | 58.4(+0.8)       | 66.6(+0.9)       | 68.0(-0.7)        | 55.7(+0.6)       | 40.7(+2.1)       | 58.7(+0.4)      |
>
> These methods often have better feature representations. **Taking the newly added NCL as an example, it integrates multiple feature learning methods and requires RandAugment as training data augmentation, making its training complexity and unfairness far exceed our method. Our method requires neither complex data augmentation nor contrastive learning.**
>
> However, these methods do not perform well in classifier learning. On the smaller CIFAR100 dataset, which is easier to fit, our method significantly outperforms these approaches. When dealing with larger datasets, where feature space plays a more crucial role, **we use our method as a Plugin to adjust the Classifier based on these method backbone, achieving performance improvements** and thus proving our method's effectiveness. Additionally, as shown in Tab.4 of the main paper, our method achieved the best improvement even with poorer feature representations, demonstrating its universality.

---

> ### Author Response · Authors · 2024-11-19
> **Response to Reviewer sTG2 [2/2]**
>
> >***Q4: The references are outdated. No papers from 2024, only 3 papers from 2023, and 4 papers from 2022.***
>
> **R4:** The main reason is that our method focuses on fundamental long-tailed distribution optimization without incorporating more complex training paradigms or introducing complicated data augmentation. **Therefore, the methods that can be fairly compared are relatively classic.**
>
> Recent works are mostly combinations of complex and effective paradigms, such as those based on mixture of experts [1], contrastive learning [2][6], data augmentation [1][3][6], and pre-trained model tuning [4][5]. While incorporating these methods can certainly improve model performance, we focus more on fundamental components rather than using an all-encompassing approach for combination. **Additionally, Q3 demonstrates that our method can further improve the performance of these methods, thereby enhancing their biased classifiers.**
>
> **Of course, in the revised version, we will actively include these methods and more in the Related Work analysis, and add the methods from Q3 to the experimental comparisons. We welcome your suggestions for references and relevant literature that should be included in our article revisions.**
>
>
>
> >***Q5: Pay attention to the format of the references.***
>
> **R5:** Thank you for your valuable feedback. During the initial submission, we relied solely on the BibTeX entries provided by Google Scholar without further refinement. Following your suggestions, we have now standardized all citation formats and updated the references from arXiv preprints to their corresponding published versions in journals and conference proceedings where applicable. **The revised PDF with these modifications has been uploaded for your review.**
>
> [1] Nested Collaborative Learning for Long-Tailed Visual Recognition. CVPR22
>
> [2] Long-Tailed Recognition by Mutual Information Maximization between Latent Features and Ground-Truth Labels. ICML23
>
> [3] Kill Two Birds with One Stone: Rethinking Data Augmentation for Deep Long-tailed Learning. ICLR24
>
> [4] Parameter-Efficient Complementary Expert Learning for Long-Tailed Visual Recognition. ACM MM24
>
> [5] Long-Tail Learning with Foundation Model: Heavy Fine-Tuning Hurts. ICML24
>
> [6] Probabilistic Contrastive Learning for Long-Tailed Visual Recognition. TPAMI24

---

> > ### Comment · Reviewer_sTG2 · 2024-11-19
> >
> > Thank you for your response. Your revisions make sense, such as including more recent work and improving formatting. I will spend some days checking the rebuttal and making appropriate updates on my decision. I have one more question: why is the title on this page ("Rethinking ...: A Simple Logits Retargeting Approach") inconsistent with the title in the PDF file ("Rethinking ...: Label Over-Smooth Can Balance")?

---

> > > ### Author Response · Authors · 2024-11-19
> > >
> > > Oh!! Sorry for making a mistake! After registration, we modified the article's title to emphasize and highlight OverSmooth, but forgot to change the previously registered title! Look at our forgetfulness, hope it didn't cause you too much trouble. Please refer to the submitted PDF version, thank you for your discovery.

---

> > > > ### Comment · Reviewer_sTG2 · 2024-11-24
> > > >
> > > > Thanks for your reply. I still have some concerns after checking your rebuttal.
> > > >
> > > > - Directly plotting a Gaussian logit distribution is not rigorous and a bit misleading, because you have not provided solid evidence that it follows a Gaussian distribution.
> > > > - The formulation of the Cosine classifier in Table 1 seems incorrect. According to your formulation, the logit $g_i$ will fall into a short interval of [-1, 1]. In this case, the prediction will be significantly limited. In practice, the Cosine similarity should be divided by a temperature, i.e., $g_i=\frac{W_i^\top}{\Vert W_i^\top\Vert}\cdot\frac{f(x)}{\Vert f(x)\Vert}/\tau$, where $\tau$ is often manually set to such as 0.1, 0.05, 0.01, etc.
> > > > - The revised PDF has improved the presentation. However, according to [Call for Papers](https://iclr.cc/Conferences/2025/CallForPapers) and [Author Guide](https://iclr.cc/Conferences/2025/AuthorGuide), the paper limit is 10 pages across all stages (initial submission, rebuttal revisions, camera ready). You'd better update your manuscript to meet the rules.

---

> ### Author Response · Authors · 2024-11-24
>
> First of all, thank you for your quick response and so many constructive suggestions. With your help, we have been continuously improving the paper. Here are the specific responses to these issues.
>
> >***Q1: Directly plotting a Gaussian logit distribution is not rigorous and a bit misleading, because you have not provided solid evidence that it follows a Gaussian distribution.***
>
> **R1**: I apologize for the confusion. The simplification to a Gaussian distribution was for intuitive visualization, but **we don't need this assumption for the subsequent proofs**. Following your suggestion, we have **used the actual distribution of the original data and updated it in the paper.** Objectively speaking, the current figure is more comprehensive than before and can effectively convey our ideas. ***Thank you for your continuous suggestions that help us further optimize the paper's readability.***
>
> >***Q2: The formulation of the Cosine classifier in Table 1 seems incorrect. According to your formulation, the logit $g_i$ will fall into a short interval of $[-1, 1]$. In this case, the prediction will be significantly limited. In practice, the Cosine similarity should be divided by a temperature, i.e.,  $g_i = \frac{W_i^T}{\|W_i\|} \cdot \frac{f(x)}{\|f(x)\|} / \tau$ where $\tau$ is often manually set to such as $0.1$, $0.05$, $0.01$, etc.***
>
> **R2**: This is a great question. Setting the temperature parameter indeed makes this method more flexible and adaptable. Based on your suggestion, we conducted urgent experiments, and here are the results - we have already updated the optimal results in the paper.
>
> | Method | $\tau$ | All | Many | Medium |  Few |
> |--------|---------|---------|---------|---------| ---------|
> | Vanilla Cosine |1 | 51.66 | 77.77 | 49.60 | 23.60 |
> | | 0.1  | 51.99 | 77.52 | 50.13 | 24.36 |
> | | 0.05 | **52.20** | 77.21 | 50.52 | 24.98 |
> | | 0.01 | 52.15 | 76.46 | 51.03 | 25.11 |
>
> >***Q3: The paper limit is 10 pages across all stages (initial submission, rebuttal revisions, camera ready). You'd better update your manuscript to meet the rules.***
>
> **R3**: Thank you for the reminder. We have formatted the article according to the specified requirements.

---

> > ### Comment · Reviewer_sTG2 · 2024-11-24
> >
> > Thank you for your additional explanations! Now I believe the proposed logit magnitude re-balancing makes sense. Therefore, I think the idea of "Label over-smoothing helps" is very interesting. However, I am confused why setting $\delta=1$ still produces good results in Figure 4  In my opinion, setting $\delta=1$ means that all classes should be predicted with an equal probability. Is it reasonable?

---

> ### Author Response · Authors · 2024-11-25
>
> In Fig. 4, while the x-axis extends to $\delta=1$, the sample points in the figure do not reach this value, with the rightmost point corresponding to $\delta=0.99$. It is also explained in Lines 375-377 of the paper: "From this perspective, $\delta$ approaching 1 can yield the best results. However, our method essentially acts as an auxiliary regularization technique to mitigate biases among different class samples and assist the model in converging. The model needs to exhibit both convergence and discriminability, meaning that the ground truth labels should remain distinct from the other labels."

---

> ### Comment · Reviewer_sTG2 · 2024-11-25
>
> Thanks for your explanation. I have run your code and find that label over-smoothing can be more surprising than expected. I find that setting $\delta=1$ still yields satisfactory results (just slightly lower than $\delta=0.98$). I think it is very interesting because it indicates that even with a uniform label distribution, label smoothing in stage 2 can still re-balance the biased predictions. But I cannot realize the underlying reasons. I guess it is because the model from stage 1 already has an initial ability. Indeed, I think your analyses in Section 3.1 have not touched the fundamental reason. Nonetheless, I believe this paper provides an interesting insight that can contribute to the community and inspire more future work. I will raise my score to 6.
>
>
>
> Moreover, previous work [1] has highlighted the benefits of label smoothing in the second stage, but it is ignored and not discussed in this paper. Besides, is there any other paper that applies label smoothing for long-tailed learning?
>
>
>
> PS: I find a mistake in your code. In `main_stage1.py`, you save a deep copy of the best model (Lines 58, 74). However, when it comes to `main_stage2.py`, you load the model using its state_dict (Lines 89-90). There is also a similar mistake in `evaluate.py`. If directly run, it will cause an error. It seems that we should use the code `model = torch.load(pth)` annotated in line 92.
>
> [1] Improving calibration for long-tailed recognition.

---

> > ### Author Response · Authors · 2024-11-25
> >
> > Thank you for pointing out this very interesting phenomenon. While our analysis led us to develop this method, there may be other explanations for the underlying reasons why it works. For example, we did not consider the issue of the initial state in one-stage training in our analysis. **We will include this observation in our paper in the hope that it will inspire further work and analysis within the community.** Thank you for your valuable review comments and contributions; we will further refine and incorporate these insights.

---

> > ### Author Response · Authors · 2024-11-26
> >
> > Sorry for the late response. Due to the lack of email notifications after changes on OpenReview, we missed your follow-up question. We will now address this oversight.
> >
> >
> > >***Q1: Moreover, previous work [1] has highlighted the benefits of label smoothing in the second stage, but it is ignored and not discussed in this paper. Besides, is there any other paper that applies label smoothing for long-tailed learning?***
> >
> > ***Q1:*** Our core component shares similarities with MisLAS[1], as both appear to utilize a form of Label Smoothing. However, **MisLAS[1] merely adjust Label Smoothing** by controlling the $\delta$ proportion for each class based on the number of class instances. In contrast, our method **fundamentally adopts counterintuitive and extreme parameter settings with large $\delta$ values**. Compared to MisLAS[1], which involves numerous hyperparameters and conservative innovations, our approach offers a more novel angle of analysis. The ultimate goal is for the model to naturally treat every class equally and solve the long tailed problem.
> >
> > We will incorporate this analysis into the next paper revision. To the best of our knowledge, there should be no other relevant works, particularly regarding innovations in Label Smoothing.
> >
> > [1] Improving Calibration for Long-Tailed Recognition, CVPR, 2021

---

### Author Response · Authors · 2024-11-21
**General Response**

We express our gratitude to all reviewers for their valuable time and insightful comments. ***We appreciate the unanimous recognition of our method's effectiveness across multiple benchmarks (sTG2, g4ig, aP5D, o1sQ), the acknowledgment of our comprehensive theoretical and experimental validation (sTG2, aP5D, o1sQ), and the appreciation of our simple yet practical approach (sTG2, g4ig, o1sQ).***

The reviewers have raised several questions and constructive suggestions, which we will ***address point by point in our rebuttal***. Meanwhile, based on the reviewers' feedback, we have made  revisions to the paper. The changes include enhanced readability, expanded discussions incorporating recent literature, and additional experimental results. ***We have uploaded the revised PDF version with all modifications highlighted in blue for your reference.***

---

### Author Response · Authors · 2024-11-23
**Inquiry for post-rebuttal comments**

Dear Reviewers:

Thank you again for your wisdom and valuable comments. We have provided complete explanations for all the questions. Since the discussion period has been underway for some time, we would be glad to hear from you whether our rebuttal has addressed your concerns. Feel free to comment on our rebuttal if you have further questions and considerations.

---

### Author Response · Authors · 2024-12-03
**Summary of the Author-Reviewer Discussion Phase**

Dear PCs, SACs, and ACs,

We sincerely thank the reviewers for their time and valuable feedback. As the discussion phase concludes, we summarize the reviewers' comments and outline the improvements made during this period. The reviewers generally regarded our method as "Effective," "Simple," "Well-experimented", featuring "Comprehensive Theoretical Analysis," "Plugin-Compatible," offering "Enhanced Evaluation Metrics," having a "Clear Structure," conducting "Comprehensive Ablation Studies," and providing "Extensive Analysis Across Various Datasets."

We summarize the reviewers' concerns and our corresponding responses as follows:

| | Reviewers' Concerns | Author's Responses |
| - | - | -|
| Common Concern | The explanation for Fig.1 is not clear. | We have provided a more detailed explanation of the illustrative figures and revised them in the updated version. These enhanced descriptions facilitate a better understanding of our subsequent analysis and ensure that the figures effectively convey the intended concepts.|
|| Why not compare with/use specific methods?| In response to all methods suggested by reviewers for comparison and integration, we have conducted additional experiments and included the corresponding results in the revised manuscript. By incorporating these relevant methods into our study, we can enhance the completeness and robustness of our work. This ensures that our comparisons are comprehensive and that our proposed method is evaluated against a broader range of existing techniques.|
|| Some of the points in the paper require further clarification.| We have provided more detailed explanations for the points that were unclear and have revised the manuscript based on the reviewers' suggestions to improve overall clarity. This includes elaborating on complex concepts, refining our arguments, and ensuring that all sections are well-articulated to facilitate better understanding by the readers.  |
| Reviewer sTG2  | The references are outdated and the format of the references needs improvement. | We have promptly supplemented the revised manuscript with a substantial amount of the latest research and analyzed that these papers focus on feature learning based on more complex paradigms, which is orthogonal to our emphasis on classifier learning methods. Therefore, we have not only included analyses of these studies but also employed our method as a plugin to adjust the classifier based on these method backbones, achieving significant performance improvements. Additionally, in the revised version, we standardized the formatting of each cited article to more clearly and uniformly display the publication conferences/journals and their dates. |
| Reviewer g4ig  | The suggested label smoothing approach doesn't directly fit into the story of achieving uniform LoMa. | We clarified our overall motivation to demonstrate how to derive the conditions that an effective method should have through our analytical process. To meet these conditions, our approach can be naturally thought of.  |
| | The math in the paper could be improved.| We believe there may have been a misunderstanding of the equation we performed. To address this, we have added more detailed explanations and comprehensive proofs in the revised manuscript. These additions are included in the appendix to provide clarity and ensure that the mathematical formulations and derivations are transparent and easily understandable. This enhancement aims to eliminate any confusion and strengthen the theoretical foundation of our work.|
|  | The proposed method is just a minor variant of prior work. | Although our proposed method is a variant of prior work, its application is entirely different. We not only introduce the concept of over-label smoothing but also provide a theoretical analysis of its impact on the model learning process. Label smoothing has not established a dominant position in the long-tailed (LT) field, and no one has attempted over-smoothing in other domains. The difference between values such as 0.99 and 0.2 goes far beyond mere formal consistency. Our method is simple yet effective and can be extremely beneficial in the LT domain, constituting a novel contribution. |
| Reviewer aP5D  | How the equations and claims is obtained? | We reiterate our initial definitions and overall reasoning to enhance understanding. By providing additional details and clarifying the processes, reviewers can better comprehend how we derived these equations and formulated our claims. |
| Reviewer o1sQ |  Recommend to include comparisons with the latest SOTA methods  and discuss related works. | We performed additional experiments and reported the results. The new experiments are included in our revised manuscript. |

---

### Public Comment · ~Yi-Hang_Zhu1 · 2025-03-03
**About reproducing the results**

(1) In the paper, appendix D, you mention that "The classifier finetune epoch is 20 with adequate learning ratio and weight." Would you please elaborate on the experimental setup for imagenet-LT and iNaturalist2018?

(2) If I understand correctly, to implement the label smoothing, I simply need to use
torch.nn.CrossEntropyLoss(label_smoothing=args.label_smooth)
during the classifier retraining, right?

(3) The label smoothing does not use the information of the number of training samples per class. Could you explain why it has the reweighting/rebalancing effect?

Thanks.

---

> ### Public Comment · ~Han_Lu2 · 2025-03-04
>
> 1. In Appendix E, we conducted parameter sensitivity experiments on CIFAR100, which demonstrated that our method is relatively stable across different parameter settings. This stability allows us to confidently apply the best-performing parameter combination from CIFAR100 to other experimental settings.
>
> 2. Indeed, your understanding is correct. We have also provided the implementation details in our GitHub.
>
> 3. Regarding this question about the reweighting/rebalancing effect, while we have provided extensive justification in the paper and previous rebuttal responses, I can offer additional intuitive perspectives:
> - As demonstrated in Figure 1, our method does not significantly alter the intra-class discriminative ability but rather reduces the majority class bias. The ultimate goal in addressing long-tailed recognition is to achieve equal treatment across all classes, which doesn't necessarily need to be accomplished through sample count manipulation.
> - In the probability prediction space, Label Over-Smooth effectively reduces the disparity in total prediction probabilities across different classes, thereby achieving the concept of equal treatment.
>
> These are preliminary insights, and we believe this phenomenon warrants various interpretations and further investigation.

---

> ### Public Comment · ~Yi-Hang_Zhu1 · 2025-03-07
>
> Dear Han,
>
> Thanks for your reply.
>
> (1) I noticed that the backbone you used for cifar100-LT is ResNet34, which provides a baseline of around 47% accuracy using your code. This is much higher than the accuracy when using ResNet32, around 38%. Most of the methods in Table 2 use ResNet32 for CIFAR100-LT, so did Alshammari et al., 2022. For a fair comparison with the existing methods, ResNet32 should be used unless you run all the methods using ResNet34.
>
> (2) When I use ResNet32 and use the common setup following the literature (Cao et al., 2019), the result I obtained when using the label smoothing is not as good as the case of not using it. I also could not reproduce your results with imagenet-LT. What do you think is the possible reason? Could you share your code for imagenet-LT and iNaturalist2018?
>
> (3) In the two-stage training from the literature, the representation part of the model is typically fixed when retraining the classifier. However, in your work, when using the label smoothing for retraining the model, you don't fix any part of the pretrained model. In the case of CIFAR100, the label smoothing 0.98 will change the labels from 1 to 0.0298 or 0 to 0.0098. Would this strong regularisation destroy the pretrained representation model? Have you tried retraining the model for, e.g., 200 epochs instead of 20?

---

### Meta-Review · Area_Chair_SoWa · 2024-12-21

**Metareview:**

This paper proposes two new metrics, Logits Magnitude and Regularized Standard Deviation, to compare the differences and similarities between various methods for classifier re-training based long-tailed recognition. Overall, the reviewers acknowledged the contribution of this paper. After rebuttal, most reviewers tend to accept this paper, yet one reviewer assumes the method as a patchwork with some takeaways. AC carefully checked the feedback and discussions and assumes the pros outweigh the cons for this work, thus AC recommends accepting this paper for publication in ICLR 2025. However, the authors should revise their final manuscript according to the valuable comments from all the reviewers.

**Additional Comments On Reviewer Discussion:**

Most of the concerns (e.g., outdated references, math issues, minor variants of prior work, etc.) are addressed by the authors. The updated version well includes the necessary details for explaining these concerns. AC agrees with the merits of this paper.

---

### Decision · Program_Chairs · 2025-01-22

Accept (Poster)